



# Measurement report: Observational Analysis of Mode-Dependent Fog Droplet Size Distribution Evolution and Improved Parameterization Using Segmented Gamma Fitting

Jingwen Zhang[1,2,3], Xiaoli Liu[1,2,3], Zhenya An[4]

[1]State Key Laboratory of Climate System Prediction and Risk Management, Nanjing University of Information Science and Technology, Nanjing, 210044, China
[2]China Meteorological Administration Aerosol-Cloud and Precipitation Key Laboratory, Nanjing University of Information Science and Technology, Nanjing, 210044, China
[3]School of Atmospheric Physics, Nanjing University of Information Science and Technology, Nanjing, 210044, China
[4]College of Meteorology and Oceanology, National University of Defense Technology, Changsha, 410000, China

*Correspondence to*: Xiaoli Liu (liuxiaoli2004y@nuist.edu.cn)

**Abstract.** Influenced by numerous physical factors, the evolution of fog droplet size distributions (DSDs) during the fog lifecycle is not yet fully understood and difficult to represent realistically in numerical models, constraining the accuracy of fog forecasting. To improve understanding of the fog evolution, field observations under a polluted background were
conducted during the winters of 2006-2009 and 2017-2018 in Nanjing, China. Among the 27 observed fog events, microphysical properties such as fog number concentration ($N_f$), liquid water content (LWC), volume-mean radius ($R_v$) and effective radius ($R_{eff}$) vary substantially. The unimodal (3 μm), bimodal (3, 21 μm) and trimodal (around 3, 13, 21 μm) DSD were observed. As the fog developed, the DSDs evolved from unimodal to multimodal. The third mode centered at 13 μm in trimodal cases appeared after the other two modes, typically around the time LWC reached its maximum, corresponding to the
mature stage of fog. For all mode types, the probability density function decreased with increasing $N_f$ and LWC. $R_v$ is generally greater than 4 μm and $R_{eff}$ greater than 6 μm for trimodal DSDs. Based on the observational findings, a segmented gamma fitting was applied to the mean DSD with partition points at 10 and 21 μm. Comparison between microphysical parameters derived from the fitted DSD and those from observations indicates that the three-segment fitting provides more accurate estimates of $N_f$ and LWC. Moreover, the three-segment gamma fitting substantially improves the representation of
$R_{eff}$, absorption coefficient and optical thickness, with most deviations constrained within 20%.

## 1 Introduction

Composed of suspended small water droplets in the air near the surface, fog has multiple impacts ranging from transportation, vegetation, air quality, human health and economy (Jia et al., 2019; Lakra and Avishek, 2022). Due the sharp decline in visibility, long duration, wide spatial coverage associated with fog, it is essential and urgent to improve fog modeling and
forecasting. The formation and evolution of fog is driven by macro and microphysical processes including radiation, turbulent,



aerosol activation and condensation (Mazoyer et al., 2022; Shao et al., 2023; Wang et al., 2020). With different physical processes interact with each other nonlinearly, fog remains as a challenging problem for numerical weather prediction (NWP), even though progress have been made in recent years (Boutle et al., 2018; Martinet et al., 2020; Tudor, 2010).

In order to gain a better understanding in mechanisms of fog evolution, in situ observations have been conducted worldwide with different regions and aerosol backgrounds (Elias et al., 2009; Gultepe et al., 2007, 2009; Haeffelin et al., 2010; Mazoyer et al., 2022). Marked variabilities of microphysical parameters such as fog number concentration ($N_f$) and liquid water content (LWC) have been found. Fogs in polluted area show higher $N_f$ due to higher aerosol number concentration ($N_a$), while in relative clean regions including mountains, rainforests and rural areas there are more big droplets, which contribute to LWC significantly (Gultepe and Milbrandt, 2010; Guo et al., 2015; Li et al., 2017; Nelli et al., 2024). Also, the fierce competition

for water vapor associated with higher $N_a$ suppresses the condensation growth in urban areas, resulting a lower $N_f$ of large droplets and smaller dispersion ($\varepsilon$) in urban fog compared to clean regions (Ge et al., 2024).

The fog droplet size distribution is a key characteristic of fog microphysical processes (Niu et al., 2012), which is influenced by aerosol chemical composition and number concentration as well as various environment factors such as temperature, humidity, wind speed and direction (Mazoyer et al., 2017; Price, 2019). The peak of the droplet number concentration spectrum

in urban fog occurs at 3 μm, compared 3.5 μm in rural areas and 4.5 μm in rainforest regions. This suggests that aerosol particles serving as cloud condensation nuclei (CCN) in rural and rainforest areas are generally larger in size compared to those in urban environments (Ge et al., 2024). Mazoyer et al. (2022) observed that fog droplet size distribution (DSD) in a semi-urban area of Paris exhibited both single mode (about 11 μm) and double mode (about 11 and 22 μm). When the fog DSD is bimodal, there is a mass transfer from smaller droplets to larger droplets due to collision-coalescence process, while

sedimentation by gravity speeds up the removal of fog droplets. The initial fog DSD is influenced by environmental supersaturation and background aerosol properties. As visibility decreases and fog develops, the DSD broadens, transitioning from unimodal to multimodal. (Mazoyer et al., 2022). In situ measurements conducted in Jinan, China revealed that when visibility is greater than 500 m, fog DSD is unimodal, with a peak at 4 μm. DSD broadened and became bimodal (5 and 14 μm) as visibility drops below 500 m. Trimodal DSDs (5, 14 and 22 μm) occurred when visibility was lower than 200 m (Wang

et al., 2021a).

Integrated with in situ measurement, numerical experiment is a commonly used approach to gain a better understanding of the physical mechanism in fog. Over the past decade, numerous numerical experiments have been conducted to evaluate the fog forecasting capabilities and limitations of various mesoscale NWP models, leading to notable progress (Cui et al., 2019; Payra and Mohan, 2014). Despite WRF has showed a well performance in forecasting certain variables such as temperature and wind,

it often struggles to capture the accurate fog lifecycle (Peterka et al., 2024; Román-Cascón et al., 2016).

Fog development and characteristic are sensitive to the shape of the DSD in models (Boutle et al., 2022). A simulation of a heavy fog event in North China Plain found that effective radius of fog droplet increases nonlinearly with aerosol number concentration (Jia et al., 2019). However, since the effective radius was obtained under the assumption of a monodisperse DSD



and the dispersion effect was neglected, it may have been overestimated or underestimated due to fog-aerosol interactions
(Chen et al., 2016; Liu and Daum, 2002). Currently, fog DSDs are described using various spectral distribution functions such
as exponential, gamma or lognormal functions in bulk microphysical scheme (Kessler, 1969; Khain et al., 2015). However,
the fog DSD exhibits strong spatial and temporal variability and evolves throughout the fog lifecycle, often displaying distinct
features such as bimodality (Nelli et al., 2024; Wang et al., 2021b). Such variabilities in DSD could cause substantial deviations
from the predefined spectral distribution functions, further bringing challenges for fog parameterization (Khain et al., 2015;
Lakra and Avishek, 2022). Therefore, a more physically consistent and adaptable representation of DSD is required to improve
simulation reliability of fog evolvement.

In this study, based on the observation data of the 27 fog events obtained in Nanjing, China during the winters of 2006-2009,
2017-2018, we focus on the characteristics and evolution of DSDs, how they are associated with microphysical characteristics
and how to improve the representation of multimodal size distributions using the gamma function. The rest of the article is
organized as follows. Section 2 describes the observation site and data. Section 3 presents the results of this study, including
an overview of microphysics across 27 fog events, an analysis the fog lifecycle under different modes, the correlations between
microphysical characteristics and varying DSD modes, and a refinement of the gamma fitting approach with an evaluation of
its performance. The discussion and main conclusions are presented in Sections 4 and 5.

## 2 Data Set and Methods

The field campaign was conducted during the winters of 2006-2009 and 2017-2018, with each campaign lasting approximately
one month per year. The sampling site was located at Pancheng (32.2 °N, 118.7 °E; 22 m above sea level), Jiangsu Province,
China, north of the Yangtze River and was surrounded by industrial facilities, residential areas, and busy roads (Niu et al.,
2010, 2012). The DSD was measured with a fog monitor (FM-100) from Droplet Measurement Technologies (DMT, USA)
with diameters ranging from 0 to 50 μm into 20 bins, at a sampling frequency of 1 Hz. The width of each bin is 2 μm for the
first 10 bins and 3 μm for the last 10 bins. To exclude the influence of large unactivated aerosol particles, data from the first
bin (0-2 μm) are omitted. Fog with $N_f > 10$ cm$^{-3}$ and $LWC > 10^{-3}$ g m$^{-3}$ was identified (Lu et al., 2020; Wang et al., 2021b).
Microphysical characteristics including fog number concentration ($N_f$), liquid water content (LWC), volume-mean radius ($R_v$),
effective radius ($R_{eff}$), dispersion ($\varepsilon$), autoconversion threshold ($T$) and first bin strength ($FBS$) were calculated through
following formulas:

$$N_f = \sum n(r) \tag{1}$$

$$LWC = 1 \times 10^{-6} \rho \sum \frac{4\pi}{3} r^3 n(r) \tag{2}$$

$$R_v = \left(\frac{\sum n(r) r^3}{N_f}\right)^{\frac{1}{3}} \tag{3}$$



$$R_{eff} = \frac{\sum n(r)r^3}{\sum n(r)r^2} \tag{4}$$

$$\sigma = (\frac{n(r)(r-\bar{r})^2}{N_f})^{\frac{1}{2}} \tag{5}$$

$$\varepsilon = \frac{\sigma}{\bar{r}} \tag{6}$$

$$T = \frac{P}{P_0} = [\frac{\int_{r_c}^{\infty} r^6 n(r) dr}{\int_0^{\infty} r^6 n(r) dr}][\frac{\int_{r_c}^{\infty} r^3 n(r) dr}{\int_0^{\infty} r^3 n(r) dr}] \tag{7}$$

$$FBS = \frac{N_{1st}}{N_f} \tag{8}$$

where $\rho$ is the density of water, $r$ is the fog droplet diameter of each bin, $\bar{r}$ is the mean arithmetic radius defined with $\bar{r} = \sum \frac{n(r)r}{N_f}$, $r_c$ is defined with $r_c \approx 4.09 \times 10^{-4} \beta_{con}^{\frac{1}{6}} \frac{N^{\frac{1}{6}}}{LWC^{\frac{1}{3}}}$, in which $\beta_{con} = 1.15 \times 10^{23}$, $N_{1st}$ is the number concentration of

the first bin (2-4 μm) following the exclusion of the 0-2 μm bin.

**3 Results**

**3.1 Overview of fog microphysics**

Summary statistics of $N_f$, $LWC$, $R_v$, $R_{eff}$, $\varepsilon$, $T$ and $FBS$ for each event are provided in Table A1 of Appendix. The average $N_f$, LWC, $R_v$ and $R_{eff}$ vary over the ranges of 25-587 cm⁻³, 0-0.28 g m⁻³, 1.6-6 μm, 1.9-8.3 μm, respectively, which shows

greater $N_f$, lower LWC and smaller droplet sizes comparing to semi-urban area in Paris, France (Mazoyer et al., 2022) and rainforest area in Xishuangbanna, China (Wang et al., 2021b). In the meantime, significant variability in the microphysical properties is observed between different events. The number of modes for each event was determined based on the presence of local minima in the DSD (Mazoyer et al., 2022). Among the 27 fog events, both unimodal and multimodal DSDs are observed. In some cases, a transition from unimodal to multimodal occurs as fog develops. Unimodal (3 μm), bimodal (3, 21

μm) and trimodal (around 3, 13, 21 μm) DSD account for 15%, 52%, and 33% of the 27 cases, respectively. Trimodal cases show greater $N_f$, LWC with bigger droplets and lower FBS. The average $R_v$ and $R_{eff}$ in trimodal cases are approximately twice those in unimodal cases, accompanied by a significantly higher autoconversion threshold ($T$). A high dispersion of 0.82 also indicates a broader DSD in trimodal events.

**Table 1 Microphysical characteristics of single mode, double mode and triple mode fog events, the first row shows the mean values, in the parentheses are the 25th and 75th percentiles of each characteristic**

| Type | Peak Diameter (μm) | $N_f$(cm⁻³) | $LWC$(g m⁻³) | $R_v$(μm) | $R_{eff}$(μm) | $T$ | $\varepsilon$ | $FBS$(%) |
|------|------|------|------|------|------|------|------|------|





| | | | | | | | | |
|---|---|---|---|---|---|---|---|---|
| Single-mode | 3 | 168.95 (45.14, 240.22) | 0.03 (0.003, 0.03) | 2.91 (2.23, 3.44) | 4.46 (2.99, 5.31) | 0.02 (0, 0.01) | 0.56 (0.43, 0.67) | 67.49 (56.99, 78.54) |
| Double-mode | 3, 21 | 235.52 (84.25, 327.64) | 0.11 (0.01, 0.15) | 4.05 (3.17, 4.84) | 6.55 (5.07, 7.93) | 0.12 (0.01, 0.15) | 0.72 (0.63, 0.81) | 56.56 (44.92, 67.91) |
| Triple-mode | 3, 11/13/15, 21 | 290.70 (192.03, 383.86) | 0.23 (0.11, 0.34) | 5.53 (4.90, 6.21) | 8.43 (7.43, 9.47) | 0.25 (0.05, 0.43) | 0.82 (0.78, 0.86) | 44.05 (36.41, 51.47) |

Figure 1 shows the mean DSDs for fog events with different modes, as well as the overall mean DSD across all cases. The overall mean spectrum lies between the mean DSDs of bimodal and trimodal events as expected. The distributions exhibit
similar $N_f$ at the small-droplet end (around 3 μm). However, as droplet size increases, the unimodal spectrum shows a sharp decrease. As aerosol activation is governed by environmental supersaturation and the aerosols hygroscopic properties (Shen et al., 2018; Wang et al., 2019), the similar $N_f$ at the small-droplet end may suggest that aerosols continue to activate throughout fog development in both unimodal and multimodal cases. Note that the mean DSD of trimodal events shows higher $N_f$ than that of bimodal cases in the 10-35 μm range, but lower $N_f$ beyond 35 μm. This raises three questions: What are the
differences in lifecycle characteristics among fog events with varying numbers of modes? How does the DSD evolve throughout the stages of fog formation, development, and dissipation? What is the relationship between the number of modes and microphysical parameters?

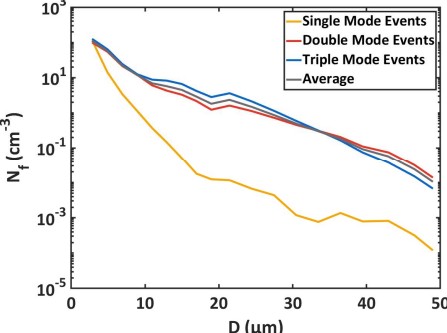

**Figure 1 Average spectrums of fog events with different modes**

### 3.2 Mode Transitions and Possible Mechanisms

To address these questions and further investigate the fog lifecycle as well as underlying physical mechanisms of different modes exhibit in DSDs, four representative cases were selected to analyze the temporal evolution of microphysical properties.



Case 6 is a unimodal event with a short duration and discontinuity in time. Case 12 is a bimodal event with a distinct formation-dissipation lifecycle lasting for approximately 2.5 hours. Cases 1-1 and 10 are both trimodal, lasting over 12 hours but displaying different lifecycle characteristics. The maximum LWC used as an indicator of mature phase is marked in Figure 2. For Fog Case 6 (F6), $N_f$ and LWC vary intensely and discontinuously at 1-minute resolution, while FBS remains relatively high. The unimodal DSDs in F6 is relatively narrow, with correspondingly small $R_v$ and $R_{eff}$, indicating that condensational

growth was limited after fog droplet activation.

Fog Case 12 (F12) shows a clear formation-dissipation evolution: before LWC reaches its maximum, $N_f$ and LWC continue to increase, followed by a gradual decrease after maximum LWC. $R_v$, $R_{eff}$ and dispersion are positively correlated with LWC, while FBS shows a negative correlation. The DSD become bimodal (3,21 μm) in only 15 minutes after fog formation. As fog develops, $N_f$ across all droplet sizes increases rapidly while the DSDs remain bimodal, indicating active condensational

growth of droplets. After reaching the maximum LWC, $N_f$ decreases accompanied by a mode transition from bimodal to unimodal as the fog dissipates. This may suggest that droplet activation/condensation and deactivation/evaporation occur simultaneously during fog evolution.

The average $N_f$ and LWC of F6 and F12 are 167.97, 168.93 cm$^{-3}$ and 0.01, 0.02 g m$^{-3}$ respectively, which is comparable. However, the DSD is initially unimodal and remains unimodal throughout fog development, whereas F12 exhibits a broader,

initially bimodal distribution. The observation site is surrounded by industrial facilities, leading to pronounced diurnal variation in aerosol concentration. Since the initial DSD is primarily affected by environmental supersaturation and aerosol chemical properties (Mazoyer et al., 2022), this may indicate that because F12 formed around midday, it was influenced by industrial emissions, resulting in the presence of more hygroscopic aerosols serving as CCN. Meanwhile, such a polluted background likely introduces larger aerosols and a broader aerosol size distribution, resulting in a wider initial DSD.

Fog Case 1 persisted for approximately 39 hours, to enable a clearer and more detailed analysis of its microphysical characteristics, the event was divided into two events at approximately the 14th hour after fog formation, based on the temporal evolution of $N_f$ and LWC. The exact initial and end times of these two events, as well as their positions within the full fog lifecycle (F1), are provided in Table A1 and Figure A1 of the Appendix.

The $N_f$ and LWC of F1-1 are 351.56 cm$^{-3}$ and 0.27 g m$^{-3}$ respectively, significantly higher than those of F10 (231.02 cm$^{-3}$ and

0.06 g m$^{-3}$), which may be attributed to differences in their lifecycle characteristics. F1-1 experienced a rapid development after formation, with $N_f$ sharply increasing to approximately 600 cm$^{-3}$ within one hour. Following the drastic intensification there is a relatively stable phase lasting about 11 hours before gradual dissipation, during which the $N_f$ and LWC remain relatively high. Fog Case 10 (F10) maintains stable with relatively low $N_f$ and LWC for the first 6 hours then experiences several pronounced fluctuations before dissipation. This difference in lifecycle also leads to distinct DSD evolution. In F10,

the $N_f$ across all sizes increases gradually and the DSDs becomes trimodal less than one hour before the time LWC reaches its maximum. Subsequently, due to significant fluctuations in $N_f$, the DSD varies accordingly, transitioning from trimodal to



bimodal and eventually to unimodal. In contrast, the rapid developments in F1-1 results in a sharp increase in $N_f$, with DSDs becoming trimodal directly from unimodal within 2 hours after fog formation, instead of experiencing a bimodal stage like F10.


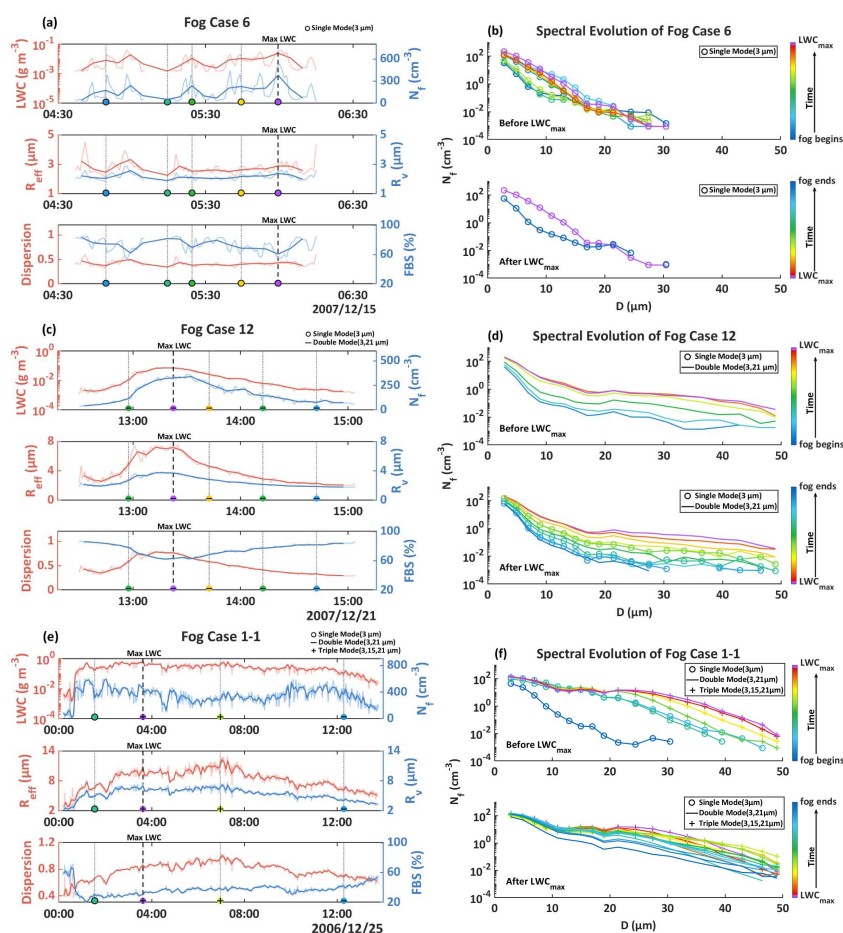



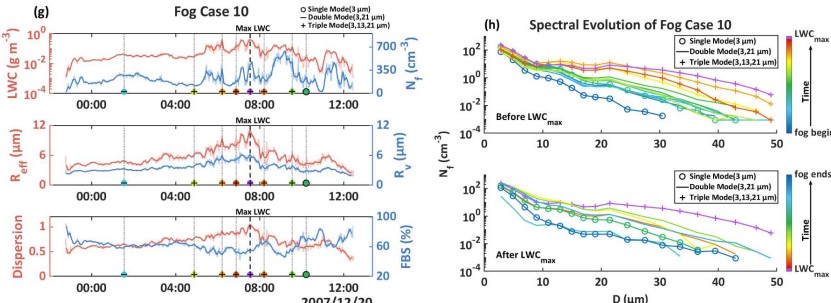

**Figure 2 (a), (c), (e), (g) are the temporal evolution of N_f, LWC, R_v, R_eff, FBS and dispersion for fog case 6, 12, 1-1, 10, the dark lines represent 5-minute averaged values while the light lines are 1-minute averaged values. (b), (d), (f), (h) are the 5-minute average DSD for case 6, 12, 1-1, 10. As LWC reaches its maximum, the colors vary from blue to red in (b), (d), (f), (h). The DSDs in (b), (d) are plotted every 10 minutes, while those in (f), (h) are plotted every 40 minutes. A part of DSDs is marked by colored dots in (a), (c), (e), (g). The positions of dots show time of each DSD and colors correspond to those in (b), (d), (f), (h).**


Note that around the time LWC reaches its maximum, both F1-1 and F10 exhibit a decrease in $N_f$ within the 10-30 μm range and an increase above 30 μm. During this period, increases in $R_v$ and $R_{eff}$ are observed in both cases, suggesting a mass transfer from smaller to larger droplets as the DSD broadens toward larger droplets. However, a slight increase in total $N_f$ and FBS is also noted simultaneously. This may suggest that aerosol continues to be activated during condensational growth,
leading to the formation of new small droplets.

A comparison between F12 and F10 reveals that the mode centered at 21 μm forms nearly simultaneously with the 3 μm mode, which is in agreement with findings from a semi-urban area of Paris (Mazoyer et al., 2022). Meanwhile, the third mode, centered around 13 μm, emerges only during the later stage of fog development, typically near the time LWC reaches its maximum, corresponding to the mature phase of the fog. This suggests that the mode centered at 13 μm likely requires
sustained water vapor supply and supersaturation, whereas the other two modes are more strongly influenced by initial environmental supersaturation and aerosol properties.

### 3.3 Correlation of modes and microphysical properties

Figure 3 presents the probability distributions of $N_f$ and LWC across unimodal and multimodal DSDs. For all modal types, $N_f$ are primarily below 750 cm^-3, with probability density functions (PDFs) decreasing as $N_f$ increases. The PDF decreases
with increasing LWC, except for the trimodal case showing an increasing trend when LWC < 0.1 g m^-3. The PDF of unimodal DSDs exhibit the strongest decline and the narrowest distribution range. In the trimodal DSDs, LWC is more likely to reach higher than 0.2 g m^-3, while the PDF in the LWC < 0.2 g m^-3 range is lower than those of unimodal and bimodal DSDs. For $R_v$ and $R_{eff}$, unimodal DSDs are mainly associated with 2-4 μm and 2-6 μm ranges respectively. In bimodal DSDs, these ranges





shift to 2-6 μm for $R_v$ and 4-10 μm for $R_{eff}$. $R_v$ is generally greater than 4 μm while $R_{eff}$ is greater than 6 μm in trimodal

DSDs.

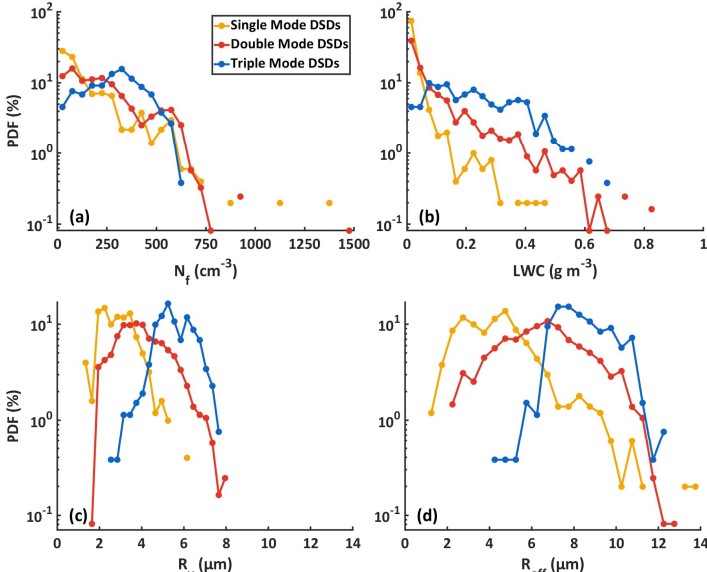

**Figure 3 PDF distributions of N$_f$ (a), LWC (b), R$_v$(c) and R$_{eff}$ (d) for DSDs with different modes**


Figure 4 shows the contributions of each droplet size bin to total $N_f$ and LWC with different modal. The contribution of each bin to $N_f$ aligns well with the number of modes. For unimodal DSDs, contributions to $N_f$ decrease monotonically with increasing droplet diameter. Compared to unimodal DSDs, bimodal and trimodal DSDs exhibit smaller FBS and greater contributions at the droplet sizes corresponding to their peaks, which are 20-23 μm for bimodal, 12-14 μm and 20-23 μm for

trimodal DSDs. However, for unimodal DSDs, the droplet size bin contributing most to LWC is not the first one, despite its accounting for 55% of the total $N_f$, indicating that larger droplets contribute more significantly to LWC. In bimodal and trimodal DSDs, contributions of each bin to LWC first increase and then decrease with droplet diameter. The greatest contribution occurs at the 20-23 μm bin, which coincides with the second mode (21 μm). In trimodal DSDs, LWC is primarily contributed by droplets in the 12-32 μm range, showing a more concentrated distribution than in bimodal cases, likely due to

the presence of the third mode around 11-15 μm.



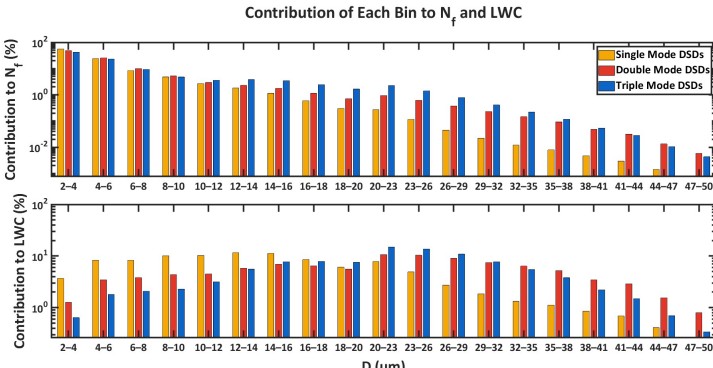

**Figure 4 Contributions of each DSD bin to N_f and LWC with different modes**

### 3.4 Performances of gamma fitting and improvement

In bulk microphysical schemes, the gamma distribution is commonly used to represent DSDs, making the accuracy of this representation critical to simulation performance. To evaluate the validity of the gamma distribution for winter fog in Nanjing, mean DSDs of 31 observed fog events are fitted by

$$n(D) = N_0 D^\mu e^{-\lambda D} \tag{9}$$

where $D$ is the droplet diameter, $n(D)$ is the number concentration for each bin, $N_0$, $\mu$ and $\lambda$ are three distribution parameters. The fitting results are shown in Figure 5.

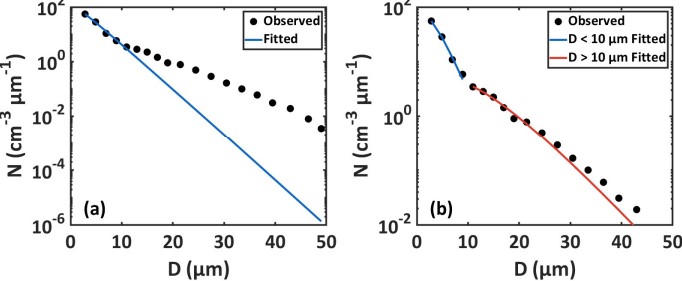





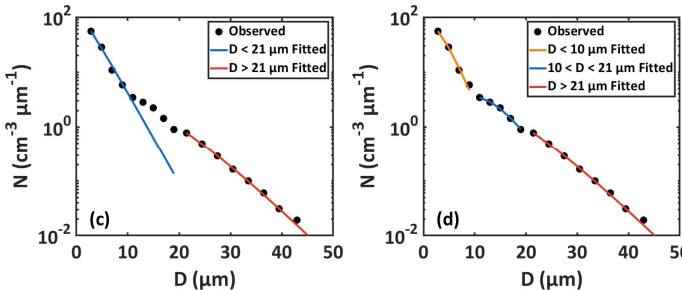

**Figure 5 Gamma fitting of the mean spectrum: original fit (a), two-segment fitting with a breakpoint at 10 μm (b), two-segment fitting with a breakpoint at 21μm (c), and three-segment fitting with breakpoints at 10 and 21 μm (d).**

For the fog events examined in this study, the gamma distribution provides a good fit to the average DSD in the small-droplet range (2-10 μm), but significantly underestimates number concentrations as droplet size increases. In multimodal DSDs of examined events, besides the peak near 3 μm, additional peaks are observed around 10 μm and 21 μm, which likely contribute to the poor fit. To address this, segmented gamma fitting was conducted using 10 μm and 21 μm as partition points. When DSDs are segmented at 10 μm, the 2-10 μm range is well represented, but the fit increasingly underestimates number concentrations for diameters above 30 μm. When segmented at 21 μm, good agreement is achieved in the 2-10 μm and 20-50 μm ranges, but the 10-20 μm segment shows substantial deviation between the fitted and observed DSD.

Based on these results, a three-segment gamma fitting approach was applied using 10 μm and 21 μm as partition points (Figure 5d). This approach significantly improves the overall fit ranging from 2 to 50 μm, providing a more accurate representation of the whole DSDs.

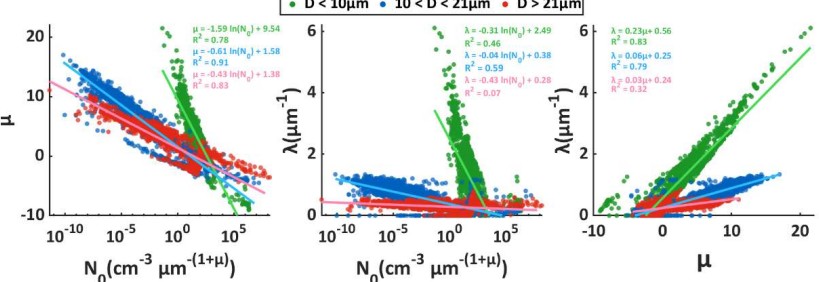

**Figure 6 Correlation between the $N_0$, μ and λ derived from the three-segment gamma fitting**

The interrelationships among the three fitting parameters obtained from the three-segment gamma fitting are shown in Figure 6. For each segment, $N_0$ exhibits a negative correlation with both $\mu$ and $\lambda$, while $\mu$ and $\lambda$ are positively correlated. In the D



< 10 μm segment, larger $N_0$ and $\lambda$ are observed, indicating a narrower and more concentrated spectrum with a steep decline

in number concentration as droplet size increases. For D > 10 μm, $N_0$ shows a wider distribution with smaller $\mu$ and $\lambda$, reflecting broader spectrum and higher number concentrations of larger droplets.

Compared to the D > 10 μm segments, $N_0$ in the D < 10 μm range is more tightly gathered in a range of $10^0$-$10^5$ cm$^{-3}$ μm$^{-(1+\mu)}$, suggesting that small droplet concentrations are higher and more consistent across different DSDs. This may indicate that environmental supersaturation is consistently sufficient to activate aerosols. However, the continued growth of activated

droplets via condensation depends on the persistence of supersaturation under vapor consumption.

To further evaluate the performance of the gamma fitting with different breaking points, $N_f$, LWC, $R_v$, and $R_{eff}$ were calculated based on both the original fit and segmented fit, and compared with those derived from observations. The correlations between fitted and observed results are analyzed in Figure 7. As shown, the non-segmented gamma fit significantly underestimates the $N_f$ of droplets larger than 10 μm, leading to underestimation of all derived microphysical quantities. These

values are scattered around the 1:1 line, indicating poor agreement with observations. The $N_f$ calculated from the two-segment fit with a breakpoint at 10 μm is close to observation, but the derived LWC and $R_v$ are slightly overestimated. For the fit segmented at 21 μm both $N_f$ and LWC are underestimated, likely due to underrepresentation of droplet concentrations in the 10-21 μm range. Compared to the non-segmented gamma fit approach, the three-segment fitting shows substantial improvement in the high $N_f$ regime ($N_f$ > 700 cm$^{-3}$) and in LWC estimation as the results converge tightly around the 1:1

line. Also, the fitting accuracy for $R_{eff}$ and $R_v$ is also improved.

Except for the non-segmented gamma fitting, the other three segmented approaches exhibit a clear pattern in estimating $R_{eff}$: underestimation primarily occurs when the observed $R_{eff}$ < 6 μm, while overestimation tends to occur when the observed $R_{eff}$ > 6 μm. This pattern is particularly pronounced in the fitting segmented at 21 μm, whereas the three-segment fitting shows notable improvement in reducing underestimation at lower $R_{eff}$ values.

Cloud optical thickness ($\tau$) and single-scattering albedo ($\omega_0$) are key parameters for evaluating the Twomey effect (Stephens, 1984; Twomey and Bohren, 1980). $\tau$ can be calculated with

$$\tau = \int \frac{3LWC}{D_{eff}} dz \tag{10}$$

where $z$ is the thickness of the cloud or fog layer (Stephens, 1978). When assuming cloud or fog is vertically homogeneous, Eq. (9) can be simplified as

$$\frac{\tau}{z} = \frac{3L}{D_{eff}} \tag{11}$$

where $\frac{\tau}{z}$ represents the average optical thickness per layer. The single-scattering albedo ($\omega_0$) can be expressed by

$$1 - \omega_0 = 1.7 k_w R_{eff} \tag{12}$$

where $k_w$ is the complex part of the refractive index of water. Eq. (11) indicates the critical role of $R_{eff}$ in the Twomey effect (Wang et al., 2019).






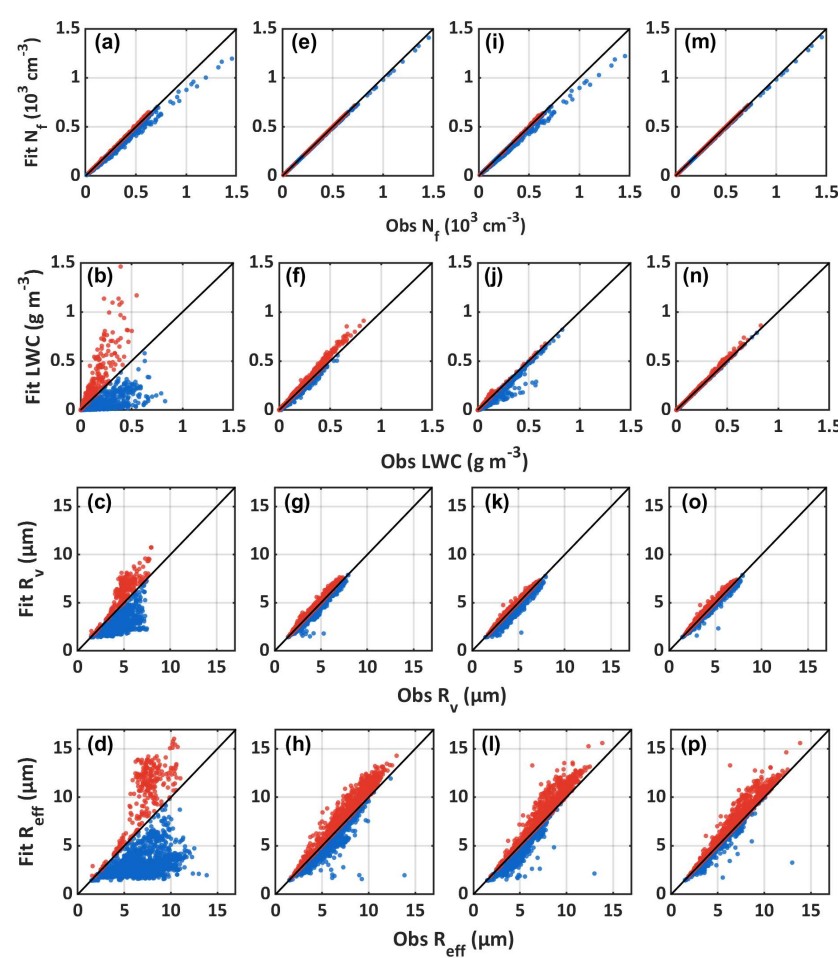

**Figure 7 Correlation between $N_f$, LWC, $R_v$ and $R_{eff}$ derived from observed spectrum and those computed from the gamma-fitted spectrum with no breakpoint (a-d), breakpoint at 10 µm (e-h), breakpoint at 21 µm (i-l) and breakpoint at 10 µm and 21 µm (m-p). Red dots indicate overestimation of the microphysical properties by the gamma fit, while blue dots indicate underestimation.**

To more precisely assess the potential climate impact of inaccuracy in $R_{eff}$ estimates from gamma fitting, Eq. (11) and (12)

were used to calculate absorption coefficient and optical thickness based on both observed and fitted $R_{eff}$. This allowed evaluation of the extent to which gamma fitting overestimates or underestimates these parameters. Results are showed in Figure 8.



Due to its significant underestimation of $N_f$ above 10 µm, the non-segmented gamma fitting notably underestimates absorption coefficient $(1 - \omega_0)$ and optical thickness $(\tau)$ by up to nearly 90%. Compared to the gamma fitting segmented at

21 µm, the fitting segmented at 10 µm more accurately captures absorption coefficient and optical thickness, yet still exhibits up to 50% overestimation or underestimation of absorption coefficient. It is noteworthy that the 21 µm-segmented fitting generally underestimates optical thickness, likely due to the underestimation of $N_f$ for droplets in the 10-21 µm range.

In contrast, the three-segment gamma fitting significantly improves the estimation of both absorption coefficient and optical thickness, with most deviations confined within ±20%. The most notable improvements lie in reducing the underestimation

of absorption coefficient and the overestimation of optical thickness.

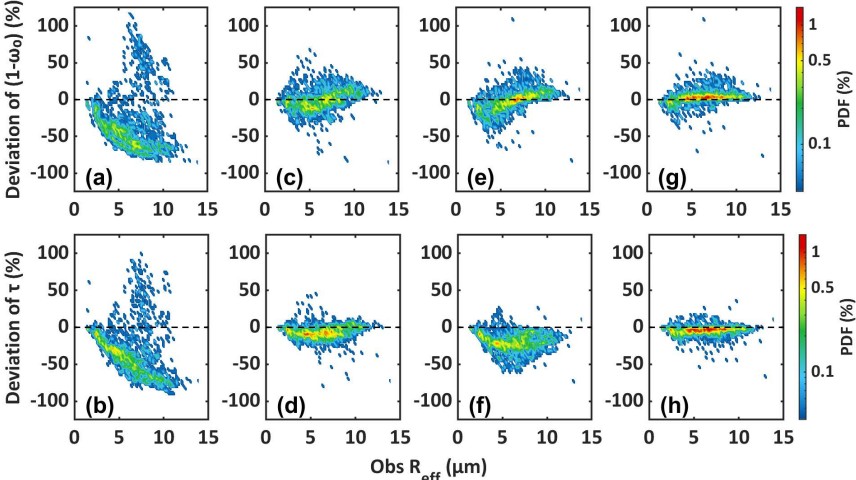

**Figure 8 Correlation between absorption coefficient $(1 - \omega_0)$ and optical thickness $(\tau)$ derived from observed spectrum and those**
**computed from the gamma-fitted spectrum with no breakpoint (a, b), breakpoint at 10 µm (c, d), breakpoint at 21 µm (e, f) and breakpoint at 10 µm and 21 µm (g, h).**

## 4 Discussion

Numerous studies on fogs in Nanjing have primarily examined the influence of meteorological factors such as temperature,
humidity, wind speed, and turbulence on fog evolution, investigated the correlations between microphysical properties, or compared microphysical properties of fogs in Nanjing with those in other regions (Ge et al., 2024; Liu et al., 2011; Lu et al., 2013; Niu et al., 2012), enhancing the scientific understanding of fog formation and dissipation in polluted urban backgrounds. However, current numerical models still struggle to accurately capture the evolution of fog lifecycles, partly due to the



inadequate parameterization of fog DSDs. In bulk microphysical schemes, DSDs are described using predefined functions
such as gamma distributions. However, previous studies have shown that DSDs vary significantly in time and space, deviating
from the idealized functions (Khain et al., 2015; Nelli et al., 2024; Wang et al., 2021b).

Based on the analysis of fog droplet size distributions (2-50 μm) obtained from 27 winter fog events in 2006-2009 and 2017-
2018 of Nanjing, China, unimodal, bimodal, and trimodal DSDs were identified across different fog events, as well as
transitions among these modes within individual cases during fog development, stabilization, and dissipation. The three
observed DSD modes peaking at 3 μm, 11-15 μm, and 21 μm are consistent with findings from both relatively clean
background in Paris, France (Mazoyer et al., 2022) and polluted region in Jinan, China (Wang et al., 2021a). Due to the poor
performance of gamma distribution in representing observed mean DSDs, segmented gamma fitting was conducted using 10
μm and 21 μm as partition points. The microphysical parameters derived from the segmented fits show markedly better
agreement with observations, with particularly notable improvements from the three-segment fitting in the high regime of $N_f$
and in LWC. Although the three-segment fitting still tends to underestimate $R_{eff}$ when $R_{eff} < 6$ μm and overestimate it
when $R_{eff} > 6$ μm, the deviations in retrieved $R_{eff}$, absorption coefficient and optical depth decreased substantially, from
values up to 90% to around 20%.

## 5 Conclusions

As a key parameter of fog microphysical processes, the droplet size distribution (DSD) is influenced by multiple macro- and
micro-scale factors, exhibits significant temporal and spatial variability, and evolves throughout the fog lifecycle, thereby
posing challenges for accurate fog prediction (Niu et al., 2012; Nelli et al., 2024). Recent study has shown fog sensitivity to
the shape of the DSD in models (Boutle et al., 2022). In bulk microphysical schemes, the gamma distribution is widely used
to represent DSD, making its accuracy critical for reliable simulations.

This study investigates the microphysical characteristics of 27 winter fog events in Nanjing under polluted conditions, with a
focus on the evolution of droplet size distributions (DSDs) throughout the fog lifecycle and on the application of segmented
gamma fitting to the mean DSD for improved parameterization. The average $N_f$, LWC, $R_v$ and $R_{eff}$ vary over the ranges of
25-587 cm$^{-3}$, 0-0.28 g m$^{-3}$, 1.6-6 μm, 1.9-8.3 μm respectively, which shows greater $N_f$, lower LWC and smaller droplets
comparing to other clean regions such as the tropical rainforests of southwestern China (Wang et al., 2021b). Among the 27
fog cases, DSDs with single mode (3 μm), double mode (3, 21 μm) and triple mode (approximately 3, 13, 21 μm) were observed.
The main findings are as follows:

Unimodal cases generally feature narrow DSDs with smaller $N_f$ and LWC. In bimodal and trimodal events, the DSD evolves
throughout the fog lifecycle: $N_f$ increases and DSD broadens as fog develops while DSD transitioning from multimodal to
unimodal during dissipation. The third mode (around 13 μm) typically emerges after the initial two (3, 21 μm) and appears
near the time LWC reaches its maximum, suggesting that the formation of the third mode requires sustained water vapor supply



and supersaturation. The underlying physical mechanisms and aerosol effects could be further investigated through sensitivity experiments using numerical modeling.

The probability density function (PDF) distributions of microphysical properties vary across spectral modes. For all modal types, the PDF decreases with increasing $N_f$ and LWC. Bimodal and multimodal DSDs are often associated with larger $R_v$ and $R_{eff}$. The contribution of each bin to $N_f$ aligns well with the appearance of modes, while larger droplets contribute

significantly to LWC.

Comparison of the retrieved physical parameters from segmented gamma fitting with observations indicates that the three-segment fitting yields the best performance, especially in improving $N_f$ and LWC estimation. Meanwhile, the three-segment fitting reduces the estimation deviations in $R_{eff}$, absorption coefficient and optical thickness from up to 90% in the non-segmented fitting to below 20%, demonstrating its effectiveness in improving fog DSD representation and microphysical

characteristic retrieval.

These findings advance our understanding of fog droplet size distribution (DSD) evolution during fog lifecycles and the correlations between DSD modes and microphysical properties, providing fundamental insights into fog microphysics in polluted urban regions such as the Yangtze River Delta. The improved segmented gamma fitting offers a new perspective for DSD parameterization and demonstrates strong potential for improving the representation of cloud/fog microphysical

processes in weather prediction and climate models.

It should also be noted that in this work, only a two-parameter gamma distribution was used to fit and refine the mean DSD. The comparative performance of alternative distribution and evaluate the influence of different parameterizations on fitting accuracy could be explored in future studies.

8000



**Appendix**

**Table A1 Initial and end times, classification, and mean microphysical properties of the 27 fog events**

| Fog Case | Initial Time (UTC+8) | End Time (UTC+8) | Type | $N_f$ (cm$^{-3}$) | LWC (g m$^{-3}$) | $R_v$ (μm) | $R_{eff}$ (μm) | T | ε | FBS (%) |
|---|---|---|---|---|---|---|---|---|---|---|
| 1-1 | 2006/12/25 00:10 | 2006/12/25 13:47 | triple mode (3,15,21μm) | 351.56 | 0.27 | 5.37 | 8.18 | 0.31 | 0.78 | 37.23 |
| 1-2 | 2006/12/25 13:48 | 2006/12/26 14:51 | double mode (3,21μm) | 349.97 | 0.23 | 4.79 | 7.67 | 0.30 | 0.75 | 42.48 |
| 2 | 2006/12/26 17:46 | 2006/12/27 01:36 | double mode (3,21μm) | 134.19 | 0.03 | 3.17 | 5.07 | 0.05 | 0.61 | 61.21 |
| 3 | 2006/12/27 03:28 | 2006/12/27 12:34 | double mode (3,21μm) | 68.47 | 0.01 | 3.32 | 5.98 | 0.04 | 0.69 | 67.33 |
| 4 | 2007/12/11 07:06 | 2007/12/11 09:55 | triple mode (3,13,21μm) | 302.68 | 0.11 | 4.00 | 6.02 | 0.01 | 0.72 | 58.31 |
| 5 | 2007/12/14 05:38 | 2007/12/14 10:20 | triple mode (3,13,21μm) | 489.08 | 0.22 | 4.51 | 6.87 | 0.12 | 0.75 | 47.73 |
| 6 | 2007/12/15 04:37 | 2007/12/15 06:15 | single mode (3μm) | 167.97 | 0.01 | 2.14 | 2.65 | 0 | 0.41 | 69.91 |
| 7 | 2007/12/15 07:06 | 2007/12/15 07:37 | single mode (3μm) | 152.81 | 0.01 | 2.05 | 2.53 | 0 | 0.39 | 73.63 |
| 8 | 2007/12/18 06:19 | 2007/12/18 10:35 | triple mode (3,13,21μm) | 199.37 | 0.05 | 3.32 | 5.39 | 0.01 | 0.66 | 68.44 |
| 9 | 2007/12/19 00:23 | 2007/12/19 11:07 | triple mode (3,13,21μm) | 254.74 | 0.08 | 3.58 | 5.47 | 0.02 | 0.69 | 63.22 |
| 10 | 2007/12/19 22:45 | 2007/12/20 12:28 | triple mode (3,13,21μm) | 231.02 | 0.06 | 3.58 | 5.54 | 0.04 | 0.69 | 62.17 |
| 11 | 2007/12/21 03:39 | 2007/12/21 08:50 | double mode (3,21μm) | 64.50 | 0 | 2.23 | 3.52 | 0.01 | 0.46 | 85.50 |
| 12 | 2007/12/21 12:30 | 2007/12/21 15:04 | double mode (3,21μm) | 168.93 | 0.02 | 2.51 | 3.97 | 0.04 | 0.50 | 75.33 |
| 13 | 2007/12/21 16:30 | 2007/12/21 17:43 | single mode (3μm) | 56.97 | 0 | 2.08 | 3.13 | 0.01 | 0.40 | 84.45 |
| 14 | 2007/12/23 01:41 | 2007/12/23 05:05 | triple mode (3,13,21μm) | 179.63 | 0.05 | 3.87 | 5.75 | 0.01 | 0.71 | 58.19 |
| 15 | 2008/12/04 21:03 | 2008/12/04 23:39 | single mode (3μm) | 154.97 | 0 | 1.62 | 1.96 | 0 | 0.19 | 98.79 |
| 16 | 2009/01/08 08:18 | 2009/01/08 12:08 | double mode (3,21μm) | 146.75 | 0.03 | 3.95 | 6.64 | 0.03 | 0.78 | 61.47 |
| 17 | 2009/12/01 20:47 | 2009/12/02 10:42 | double mode (3,21μm) | 234.55 | 0.06 | 3.67 | 6.59 | 0.09 | 0.77 | 68.74 |
| 18 | 2017/12/31 05:10 | 2017/12/31 11:00 | double mode (3,21μm) | 597.88 | 0.21 | 4.19 | 6.33 | 0.05 | 0.72 | 43.00 |
| 19 | 2018/01/05 21:24 | 2018/01/05 23:48 | triple mode (3,13,21μm) | 249.77 | 0.06 | 3.77 | 5.34 | 0 | 0.63 | 46.08 |
| 20 | 2018/01/07 08:41 | 2018/01/07 12:33 | double mode (3,21μm) | 44.20 | 0.01 | 3.55 | 6.38 | 0.02 | 0.75 | 69.38 |
| 21 | 2018/11/26 20:47 | 2018/11/27 08:57 | triple mode (3,11,21μm) | 164.68 | 0.13 | 6.02 | 8.80 | 0.22 | 0.77 | 40.53 |





| 22 | 2018/11/28 08:00 | 2018/11/28 09:22 | double mode (3,21$\mu m$) | 104.31 | 0.03 | 4.41 | 7.87 | 0.07 | 0.86 | 66.18 |
| 23 | 2018/12/01 01:10 | 2018/12/01 08:08 | double mode (3,21$\mu m$) | 107.65 | 0.01 | 2.88 | 4.71 | 0.02 | 0.61 | 71.75 |
| 24 | 2018/12/20 03:22 | 2018/12/20 08:05 | double mode (3,21$\mu m$) | 80.27 | 0.01 | 3.09 | 4.12 | 0 | 0.53 | 52.50 |
| 25 | 2019/01/04 21:49 | 2019/01/05 03:50 | double mode (3,21$\mu m$) | 48.77 | 0.01 | 2.83 | 4.75 | 0.04 | 0.56 | 72.61 |
| 26 | 2019/01/06 07:20 | 2019/01/06 08:39 | double mode (3,21$\mu m$) | 25.70 | 0.01 | 4.06 | 7.75 | 0.06 | 0.87 | 73.22 |

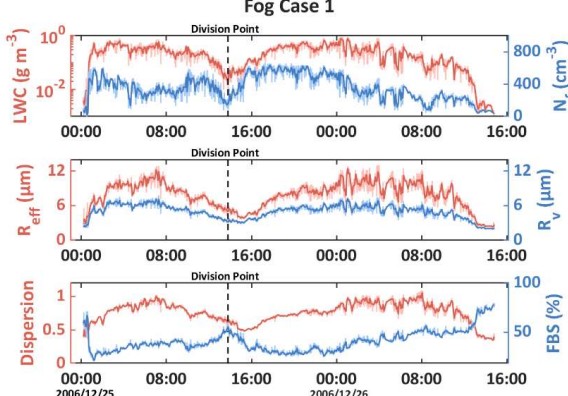

**Figure A1 the temporal evolution of $N_f$, LWC, $R_v$, $R_{eff}$, FBS and dispersion for fog case 1, the dark lines represent 5-minute averaged values while the light lines are 1-minute averaged values, the black dashed line is the boundary between F1-1 and F1-2**

**Data availability**

Data used in this study is available at Zenodo: https://doi.org/10.5281/zenodo.16883670 .

**Author contributions**

JZ and XL shaped the concept of this study and performed data analysis. JZ prepared the figures and wrote the initial draft. XL supervised this work and revised the article. ZA gave suggestions on data processing and visualization.

**Acknowledgements**

We gratefully acknowledge Professor Shengjie Niu and his research team for their efforts in acquiring and sharing the fog microphysical observation data that supported this study. We would like to acknowledge the High Performance Computing



Center of Nanjing University of Information Science and Technology and the National Key Scientific and Technological

Infrastructure project "Earth System Numerical Simulation Facility" (EarthLab) for their support of this work.

**Financial support**

This study was supported by the National Natural Science Foundation of China (GrantNos.42061134009 and 41975176) and the National Key Scientific and Technological Infrastructure project "Earth System Numerical Simulation Facility" (EarthLab).

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
