# Peer review of "Measurement report: Observational Analysis of Mode-Dependent Fog Droplet Size Distribution Evolution and Improved Parameterization Using Segmented Gamma Fitting"

_EGUsphere, 2025_

## Referee Comment (RC2)

Submitted to EGUsphere

Title: Measurement report: Observational Analysis of Mode-Dependent fog DSD Evolution.....

By Zhang et al

Date: Oct 06 2025

Decision: Major improvements are needed, if not corrected properly,  suggest its rejection.

General summary

This work uses FM100 measurements over 27 fog events and I suggest its DSD to be represented by 2-3  peak-modal distributions.  Then, provide summary of mean values of MP parameters. They represent DSD in segments and apply Gamma-DSD to show how good fits. Results are also used for evaluating optical thickness and SSA. Then, conclude that these MP parameterizations based on GDSD can improve NWP models.

Overall, its goal is acceptable scientifically. But having only DSD from FM100 measurements and do various gamma DSD fits do not qualify this paper's publishing. We all know that DSD may not follow up a single mode GDSD, and we use double or even triple mode GSDS based on segmenting the measurements. The work should do beyond the GDSD fits and I summarized below:

1.  Paper is mainly focusing on mostly Chinese studies (see discussion section, and others). Intro is better but also limited in scientific content. Earlier work were not mentioned properly.
2.  Method; provides equations but analysis is not clear to me. For example, macro and micro processes are mentioned; where are the macro processes/conditions? You have 27 events? It means what? No synoptic conditions are summarized. High pressure? Low pressure? Even no temperature range/RH?
3.  Where are the temperature, RH and wind measurements? What caused  DSD having 3-modes? Why?
4.  Where is the Vis from PWD? You have FM100, I am sure you have PWD Vis data too?
5.  Fig. 1 summarized the results; when I see this figure, I thought this paper should be rejected. Reason is that single mode is very different than others but other are almost same values with very small differences.
6.  Fig. 1 and analysis; what are the time averages used? 1 sec? 1 min , 10 mins, 60 mins and why? I don't see this. Your results are time dependent/averages. All results should be revaluated.

7. Where are the time series of Nd, LWC, and MVD for at least 2 events?
8. How come LWC becomes close to 1 g m-3? For some cases?
9. Nd is more than 200 cm-3 and goes up to 600 cm-3; where is PWD Vis comparison?
10. At least show a couple picts from camera for dense fog events?/satellite images please, 3.9 micron channels.
11. Ln27; fog definition is wrong, see Gultepe et al 2016 Gultepe et al Atmos Res; AMS Bull on ice fog. Fog can be liquid or ice ...
12. Large scale processes ; see Gultepe et al 2021 BLM
13. Ln43,44,45 ; 3, 3,5 and 4.5 micron numbers mean nothing because they are in the range of measurement uncertainty at about 1-2 micron.
14. First bin is removed (ref: Gultepe and Isaac QJRMS 200?)
15. Ln 49; how do you know because of collision processes? Where is the wind measurements in this paper?
16. Ln63; decreases I guess.
17. Ln59; T and RH, wind measurements still have very large uncertainty in predictions; who said they are correct?
18. Ln66-67; not bulk also with bin MP.
19. Ln73; what kind of fog; radiation? Advection? Marine etc.
20. Ln83; should be 1-50 micron; not 0-50 micron.
21. Method section is not clear to me, providing eqs are not enough. How did you make averages? Where is the Vis etc.
22. Table 1; show a figure with 1 sec spectra of the dsd.....
23. Ln123; why Nd is more than bimodal case? Explain it
24. 25; figure 2; make a square box for DSD same length in x and y.
25. Until fig 2; no averages mentioned; why those averages are taking?
26. Fig 3a Nf spectra, pdf are almost same; is it contradict to other events/cases etc. what is averaging time?
27. Fig06; represents what? Using obs or theretical data?
28. Eq. 11; not z but dz
29. Fig. 7; you have many points with LWC>0.5 g m-3? In polluted environment you should have that high. Why is that?
30. LN315-318; see rev paper on Fog Pure and Appl Geop 2007 Gultepe et al.
31. Ln342; I don't think Nf increases with broadening, should decrease. This contradicts everything I know for DSD. You need wind measurements.......

Overall; needs a major work, intention is good but analysis and representation are very poor. This work needs additional observations/knowledge to improve the paper. I suggest major corrections and improve t4ext flow.

---

## Author Comment (AC1)

Dear Reviewer,

We sincerely thank you for your thorough and constructive review of our manuscript. Your comments have greatly helped us identify key areas that required clarification and improvement. We have carefully revised the manuscript in response to all your suggestions. Below, we provide a point-by-point reply, detailing the changes made and the corresponding locations in the revised manuscript. All major methodological explanations, theoretical background, and figure revisions have been substantially strengthened according to your feedback.

Q1- The article focuses on DSDs with different modes—unimodal, bimodal, and trimodal. However, after reading the introduction, I am still unclear on what the three modes of a fog DSD refer to, the underlying physics driving their development, and why the modes center on the highlighted diameters. Since the study emphasizes the identification and implications of DSD modes, the authors should provide more theoretical discussion and relevant literature on fog DSDs.

Answer: Thank you for the suggestion. Following your advice, we have expanded the *Introduction* in the revised manuscript to include additional content on DSD and its multimodal characteristics to strengthen the scientific background of this work. This includes a description of unimodal, bimodal, and trimodal DSDs, findings of previous observations of multimodal DSDs, the influence of DSD on fog optical properties and radiation, and the importance of representing DSD in fog numerical forecasting. For example:

About description of unimodal, bimodal, and trimodal DSDs:

"*Fog droplet size distributions (DSDs) often exhibit one or more distinct peaks, referred to as unimodal, bimodal, or trimodal DSD, and can be attributed to different origins of the fog and processes within it (Elias et al., 2015; Hammer et al., 2014; Sampurno Bruijnzeel et al., 2005).*"

About findings of previous observations of multimodal DSDs:

"*KUNKEL (1982) finds various shapes in DSDs measured in advection fogs. Many other studies have shown the existence of bimodal DSDs in mature radiation fogs (Meyer et al., 1980; Pinnick et al., 1978; Roach et al., 1976; Wendisch et al., 1998). Gultepe and Milbrandt (2007) reported DSD modes near 4 and 23 μm during winter fog events in the Toronto region. Boudala et al. (2022) investigated the seasonal and microphysical characteristics of fog at Cold Lake Airport in northern Alberta, Canada, and found that radiation fog exhibited a bimodal droplet spectrum with peaks at 4 μm and 17-25 μm.*"

About the influence of DSD on fog optical properties and radiation:

"*Stewart and Essenwanger (1982) showed that the attenuation of electromagnetic radiation by fog depends sensitively on the shape of the droplet size distribution. The DSD and water vapor determine the overall optical properties of fog and its effects on visibility and radiative transfer together.*"

About the importance of representing DSD in fog numerical forecasting:

"*The simulated evolution of fog exhibits a sensitivity to the shape of the DSD comparable to its sensitivity to aerosol loading or cloud droplet number concentration (CDNC), however it remains one of the least investigated and rarely adjusted components of microphysical parameterization schemes(Boutle et al., 2022).*"

Q2- Most results in Sections 3.2 and 3.3 rely on identifying multimodal DSDs, yet these modes often appear to be determined by the behavior of a single size bin. This raises concerns about sampling errors, which the authors have not adequately addressed. They briefly mention using local

minima to quantify the number of PSD modes, but the cited reference appears to distinguish only between unimodal and bimodal DSDs. Furthermore, the role of time averaging is not discussed in the data section, and the fact that results are based on 5-minute averages is only noted in figure captions. A better discussion of how the multimodal DSDs are defined and distinguished is necessary.

Answer: We appreciate this important concern. In the revised manuscript, we clarified the procedure used to determine the number of DSD peak modes and their peak diameters. Each fog DSD was sequentially fitted with unimodal (i=1), bimodal (i=2), and trimodal (i=3) gamma and lognormal distributions as show below:

For the gamma distribution:

$$n(D) = \sum_{i=1}^{3} n_i(D) = \sum_{i=1}^{3} N_{0,i} D^{\mu_i} e^{-\lambda_i D}$$

For the lognormal distribution:

$$n(D) = \sum_{i=1}^{3} n_i(D) = \sum_{i=1}^{3} \frac{n_i}{2\pi^{1/2} D_i ln\sigma_{g,i}} \exp\left(-\frac{(lnD_i - lnD_{g,i})^2}{2(ln\sigma_{g,i})^2}\right)$$

By setting the derivative to zero and solving, the peak diameter ($D_{peak}$) can be expressed as a function of the fitting parameter $D_g$ and $\sigma_g$:

$$D_{peak} = D_g \exp\left(-(ln\sigma_g)^2\right)$$

We fitted the observed DSDs with unimodal, bimodal, and trimodal gamma and lognormal distributions. For each DSD, unimodal, bimodal, and trimodal fits using both the gamma and lognormal distribution were evaluated. The Akaike Information Criterion (AIC) and the Bayesian Information Criterion (BIC) were used to determine which of the three fits provides the best representation of the given DSD within each distribution type, and this fit was considered as the optimal fit. AIC provides a numerical basis for ranking competing models by their information loss in approximating the unknown true process, with the model yielding the lowest AIC considered the best approximating model (Symonds and Moussalli, 2011). BIC is consistent in the sense that it selects the true model with probability approaching one. A lower BIC corresponds to a higher posterior probability for the model and is therefore regarded as indicating a better model (Chakrabarti and Ghosh, 2011). Within each distribution type, the unimodal, bimodal, and trimodal fits are compared, and the one with the lowest AIC and BIC was considered the optimal fit of the given DSD.

We then retrieved $N_f$, $LWC$, $R_v$ and $R_{eff}$ from the optimal gamma and lognormal fits and compared them with the observations. The lognormal function provided a more accurate representation of the DSDs, so we used it to determine the number of peak modes and peak diameters in this study, and we reclassified all fog cases accordingly. The detailed formulas, calculation procedures, figures and descriptions of AIC and BIC are provided in Section 2 (*Data Set and Methods*).

We have also added a discussion on the 1-min and 5-min temporal resolutions at the end of Section 2 (*Data and Methods*) in the revised manuscript. We fitted the DSDs at both resolutions, derived the peak mode numbers and peak diameters, and reclassified all fog cases accordingly. The results show that the determination of peak mode numbers and peak diameters of DSD is not sensitive to temporal resolution, with only minor differences between the two. To reduce noise, we adopt the 5-min resolution in this study. The detailed classifications and modal diameters for both resolutions are provided in Table A2 in the appendix.

[Figure]

*Figure 2 (a) The D_{peak} distributions of all DSDs at 1-min and 5-min resolution, and (b) the D_{peak} distributions by fog type at both resolutions*

Q3- I find the interpretation of results in Section 3.2 difficult to follow. Figures 2b, d, f, and h contain overlapping DSDs in multiple colors that are not consistently referenced in the results or discussion. I recommend highlighting only the DSDs relevant to the discussion and removing or de-emphasizing distracting information.

Answer: Thank you for highlighting this readability issue. We have redesigned the figures to improve clarity. In the revised version, we selected a subset of DSDs based on the evolution of the fog life cycle. The time marked in the physical-variable time series (a) correspond exactly in number and color to the DSD spectrum (b). This mapping highlights how the DSD evolves through the fog life cycle and avoids overlap issues. For example:

[Figure]

*Figure 4 (a) is the temporal evolution of N_f, LWC, R_v, R_{eff}, FBS and dispersion for fog case 22, the dark lines represent 5-minute averaged values while the light lines are 1-minute averaged values. (b) is the 5-minute average DSD. As LWC reaches its maximum, the colors vary from blue to red, and each DSD in (b) are marked by colored dots in (a) with each color corresponding to a specific DSD and its number of peaks indicated. (c) is the temporal evolution of visibility, temperature and wind speed, with wind direction represented by wind barbs. (d) is the 1-minute temporal evolution of DSDs.*

Q4- Section 3.4 compares a triply partitioned PDF fit-implicitly tied to the three modes of fog DSDs-to that of a single gamma distribution. However, a major null hypothesis remains unaddressed: any

DSD is likely to be better represented by a partitioned PDF than a single PDF, particularly when large DSDs (which may be assigned as trimodal) are examined. The authors should test their partition points against arbitrary alternatives to demonstrate that the observed improvements are physically meaningful and not simply an artifact of partitioning.

Answer: Thank you for point out this issue. In the revised manuscript, we discuss whether the segmentation at 10 and 22 μm has physical significance. The specific analysis is as follows:

"*To demonstrate that the superior performance of the three-segment gamma and lognormal fitting is due to the physically meaningful segmentation based on the characteristics of DSDs, rather than merely the increased number of segments, we evaluated the performance of alternative segmented fitting. Since the gamma and lognormal distributions are nonlinear, two fitting points would fall on a straight line and cannot uniquely constrain the curvature of the distribution, potentially leading to non-identifiable or ill-posed parameter estimates. Therefore, the segmentation points must satisfy two conditions: the full spectrum must be divided into three segments, and each segment must contain at least three bins. Under these constraints, 66 feasible segmentation combinations exist. Using each set of segmentation points, we performed gamma and lognormal fits for all DSDs and retrieved the corresponding $N_f$, LWC, $R_v$, and $R_{eff}$. The absolute deviations between the retrieved values and the observed ones were compared for both the alternative fittings and the fixed 10 μm and 22 μm segmentation (Figure 13). For all four microphysical parameters, the deviations from the fixed-segmentation fitting are significantly smaller than those from alternative segmentation. A two-sided binomial test was conducted to evaluate the probability that the fixed segmentation outperforms the alternative segmentation. For both distributions, the 95% confidence interval is [0.986, 1.000] with a p-value of 2.23e-308. These p-values are far below the 0.05 significance threshold. These results confirm the effectiveness of the 10 μm and 22 μm segmentation for both gamma and lognormal distribution.*"

[Figure]

***Figure 13 Boxplots of the mean absolute deviations from 66 alternative segmented fittings, with red dots indicating the mean deviations from the fixed 10 μm and 22 μm segmentation***"

Q5- I am also uncertain about the gamma PDF fitting methodology used in this section. In Figure 5a, the "gamma fit" appears indistinguishable from an exponential distribution. Since the gamma DSD is inherently multimodal—limiting to a power law for small particles and an exponential form for larger particles—it is unclear why the fit reduces to a simple exponential capturing only a handful of data points. A gamma PDF with a negative μ parameter would typically capture the tail behavior

more effectively. My impression is that the authors restricted μ > 0, but no explanation of the fitting procedure or parameter constraints is provided, which is also an issue.

Answer: Thank you for this crucial point. During the gamma fitting of the mean spectrum, we set the lower bounds of $N_0$, μ, and λ to 0, −10, and 0, respectively, and did not impose μ > 0. In our fitted mean spectrum, μ is indeed negative ($-2.61 \times 10^{-7}$). In another study of winter fog in Nanjing (Wang Y, Lu C, Niu S, et al. Diverse dispersion effects and parameterization of relative dispersion in urban fog in eastern China[J]. Journal of Geophysical Research: Atmospheres, 2023, 128(6): e2022JD037514.), we also found that the gamma distribution provides a poor fit to the mean fog spectrum.

[Figure]

*Figure 9. (a) Probability density function (PDF) of the relationship between the fog droplet number concentration ($dN_f/dD$) and droplet diameter (D). The gray line is the mean fog droplet number size distribution (FDSD) and the dash line is the Gamma fitting of the FDSD.*

Q6- Some final opinions on the structure of the paper, but these aren't science related. Lines 124–127 should be research questions in the introduction, and the discussion section is unnecessary in it's current form since the two paragraphs could be moved to the introduction and conclusion.

Answer: We appreciate this structural advice. In the revised manuscript, we have deleted the discussion section and revised the conclusion section. New conclusion section as follows:

*"As a key parameter of fog microphysical processes, the droplet size distribution (DSD) is influenced by multiple macro- and micro-scale factors, exhibits significant temporal and spatial variability, and evolves throughout the fog lifecycle, thereby posing challenges for accurate fog prediction (Niu et al., 2012; Nelli et al., 2024). Recent study has shown fog sensitivity to the shape of the DSD in models (Boutle et al., 2022). In microphysical schemes of numerical model, the gamma and lognormal distribution is widely used to represent size distributions of cloud or fog droplets, making its accuracy critical for reliable simulations.*

*This study investigates the microphysical characteristics of 27 winter fog events in Nanjing under polluted conditions, with a focus on the evolution of droplet size distributions (DSDs) throughout the fog lifecycle and on the application of segmented gamma fitting to the mean DSD for improved parameterization. The average $N_f$, LWC, $R_v$ and $R_{eff}$ vary over the ranges of 25-587 cm$^{-3}$, 0-0.28 g m$^{-3}$, 1.6-6 μm, 1.9-8.3 μm respectively, which shows greater $N_f$, lower LWC and smaller droplets comparing to other clean regions such as the tropical rainforests of southwestern China (Wang et al., 2021). Among the 27 fog cases, DSDs with single mode (3 μm), double mode (3, 7-13 μm) and triple mode (3, 9-15, 21-25 μm) were observed. The main findings are as follows:*

*Among all fog cases, radiation fog accounts for the largest proportion. Radiation-advection fog tends to persist longer and is typically associated with trimodal DSDs. Unimodal cases are more likely to occur when the fog duration is short, the DSD is narrow, or the FBS is high. For bimodal and trimodal cases, both the number of peaks and their diameters vary with the fog life cycle. As the fog develops, sustained condensational growth often leads unimodal and trimodal DSDs to evolve into bimodal, with more concentrated peaks. The peak diameters are linked to the ability of fog to maintain high number concentrations and liquid water content. Although the $N_f$ of larger droplets increases, droplets around 3 μm consistently exhibit the highest $N_f$, indicating continuous activation and formation of new droplets during the condensational growth.*

*The probability density function (PDF) distributions of microphysical properties vary across spectral modes. For all modal types, the PDF decreases with increasing $N_f$ and LWC. Compared to trimodal DSDs, the PDF distributions of $R_v$ and $R_{eff}$ in bimodal DSDs are more concentrated. The contribution of each bin to $N_f$ aligns well with the appearance of modes, while larger droplets contribute significantly to LWC.*

*Comparison of the retrieved physical parameters from segmented gamma and lognormal fitting with observations indicates that the three-segment fitting yields the best performance, especially in improving $N_f$ and LWC estimation. Meanwhile, the three-segment fitting reduces the estimation deviations in $R_{eff}$, absorption coefficient and optical thickness from up to 90% in the non-segmented fitting to below 20%, demonstrating its effectiveness in improving fog DSD representation and microphysical characteristic retrieval.*

*These findings advance our understanding of fog droplet size distribution (DSD) evolution during fog lifecycles and the correlations between DSD modes and microphysical properties, providing fundamental insights into fog microphysics in polluted urban regions such as the Yangtze River Delta, China. The improved segmented gamma and lognormal fitting offers a new perspective for DSD parameterization and demonstrates strong potential for improving the representation of cloud/fog microphysical processes in weather prediction and climate models.*

*It should also be noted that in this work, only a three-parameter gamma and lognormal distribution was used to fit and refine the mean DSD. The comparative performance of alternative distribution and evaluate the influence of different parameterizations on fitting accuracy could be explored in future studies."*

---

## Author Comment (AC2)

Dear Reviewer,

We sincerely thank the reviewer for the thorough and constructive comments. Following the reviewer's suggestions, we have expanded the methodological descriptions, added missing observations, and reprocessed several figures. Below we provide detailed responses to each comment and describe all corresponding revisions made in the manuscript.

Q1- Paper is mainly focusing on mostly Chinese studies (see discussion section, and others). Intro is better but also limited in scientific content. Earlier work were not mentioned properly.

Answer: Thank you for your suggestion. In the revised manuscript, we have substantially expanded the *Introduction* by incorporating a broader range of studies worldwide, spanning from the 1970s to the 2020s, to strengthen the scientific background of this work. The updated *Introduction* now includes additional material on fog classification, the influence of large-scale processes on fog formation, early and global studies of multimodal fog DSDs, and the impact of DSD characteristics on fog optical properties. These additions provide clearer context for our research objectives and results. For example:

About fog classification:

"*Meteorological variables and large-scale processes strongly influence fog formation. In general, fog can be classified into two categories: airmass fog and frontal fog, which can be further divided into cold- and warm-advection fog, radiation fog, and sea fog etc. (WILLETT, 1928). Another fog-forming mechanism is the overall lowering of a cloud layer, including its cloud top (Koračin et al., 2014). Radiation fog typically forms near the surface under clear skies and weak winds associated with anticyclonic conditions. Its primary mechanism is radiative cooling, while opposing effects include upward soil heat flux and the warming and moisture loss caused by turbulent mixing within the stable boundary layer (Brown, 1980; Roach, 1976; Turton and Brown, 1987). The advection fog is associated with the advection of a moist air mass with a temperature contrast relative to the underlying surface, which is mainly coastal but also can be observed over land (Friedlein, 2004). Advection-radiation fog is produced by the radiative cooling of moist air that has been advected inland from the ocean or another large water body (Ryznar, 1977).*"

About large-scale process:

"*The C-FOG field campaign along the Atlantic Canada and northeastern U.S. coastlines showed that coastal fog was influenced by multiple weather systems, including northeastern high pressure, west-northwest low pressure, and tropical cyclonic activity (Gultepe et al., 2021). Another study in the region found that fog associated with cyclonic systems was consistently produced by cloud-base lowering and subsequent downward extension to the surface, whereas anticyclonic fog developed either from surface radiative cooling or from the downward extension of low-level stratus to the surface (Dorman et al., 2021).*"

About early and global studies of multimodal fog DSDs:

"*Fog droplet size distributions (DSDs) often exhibit one or more distinct peaks, referred to as unimodal, bimodal, or trimodal DSD, and can be attributed to different origins of the fog and processes within it (Elias et al., 2015; Hammer et al., 2014; Sampurno Bruijnzeel et al., 2005). KUNKEL (1982) finds various shapes in DSDs measured in advection fogs. Many other studies have shown the existence of bimodal DSDs in mature radiation fogs (Meyer et al., 1980; Pinnick et al., 1978; Roach et al., 1976; Wendisch et al., 1998). Gultepe and Milbrandt (2007) reported DSD modes near 4 and 23 μm during winter fog events in the Toronto region. Boudala et al. (2022) investigated the seasonal and microphysical characteristics of fog at Cold Lake Airport in northern Alberta, Canada, and found that radiation fog exhibited a bimodal droplet spectrum with peaks at 4 μm and 17-25 μm.*"

Q2- Method; provides equations but analysis is not clear to me. For example, macro and micro processes are mentioned; where are the macro processes/conditions? You have 27 events? It means what? No synoptic conditions are summarized. High pressure? Low pressure? Even no temperature range/RH?

Answer: We appreciate your feedback. The classifications of the 27 fog events (radiation fog, advection fog, and radiation-advection fog) have been added to Table A1 in the appendix. For the five fog cases analyzed in detail in the manuscript (fog case 1-1, fog case 10, fog case 4, fog case 22 and fog case 20), surface, 850 hPa, and 700 hPa synoptic charts derived from ERA5 reanalysis have been included in the appendix (Figures A6-A10). Weather-condition analyses for these fog events have also been added to section 3.2 *Mode Transitions and Possible Mechanisms*, for example:

*"Fog Case1 was a radiation-advection fog event lasting 39 hours. It formed under radiative cooling conditions, with sustained southwest warm and moist airflow supporting its long duration (Figure A8)."*

*"Fog Case 4 (F4) was consistent with radiation fog as the sampling site was under high-pressure control with weak surface winds, and the densely packed specific humidity contours at 850 hPa indicated ample moisture (Figure A7)."*

Q3- Where are the temperature, RH and wind measurements? What caused DSD having 3-modes? Why?

Answer: Thank you for your suggestion. Temperature and wind speed data for all fog cases have been added to Table A1 in the appendix, including the mean, 25th percentile, and 75th percentile. For the five fog events analyzed in detail (fog case 1-1, fog case 10, fog case 4, fog case 22 and fog case 20), time series of temperature, wind speed, and wind direction have been included. We have also added an analysis of the influence of meteorological variables on fog in Section 3.2 *Mode Transitions and Possible Mechanisms*, for example:

*"Fog Case 22 (F22) was a radiation-advection fog event. Under nocturnal radiative cooling on a clear night, surface temperature decreased and light fog formed around 2018/11/28 06:00. Weak warm, moist advection at 850 hPa further supported its development and persistence (Figure A6). At fog onset, visibility decreased sharply. After LWC reached its maximum, visibility and temperature gradually increased, and around 09:00 the rising wind speed and shifting wind direction accelerated the fog dissipation."*

In this study, trimodal DSDs mainly appear during the onset and dissipation stages of fog. We speculate that the trimodal DSDs observed at the onset of fog are related to the activation of aerosols with different hygroscopicity and critical supersaturation. We found that during the mature stage, especially in cases (such as fog case 1-1 and fog case 10) with high droplet number concentration and LWC, trimodal DSD which usually peaking around 3, 13, 22 μm gradually shifted to bimodal with peak diameters at 3 and 12-19 μm. This may be attributed to the convergence of droplet sizes during condensational growth.

Q4- Where is the Vis from PWD? You have FM100, I am sure you have PWD Vis data too?

Answer: Thank you for your suggestion. Visibility data for all fog cases have been added to Table A1 in the appendix, including the mean, 25th percentile, and 75th percentile. For the five fog events analyzed in detail (fog case 1-1, fog case 10, fog case 4, fog case 22 and fog case 20), visibility time series have also been included. In the revised manuscript, the *Data Availability* section has been updated to provide a detailed description of the sources of the visibility data:

*"Except for fog case 1-3 and 18-20, where visibility was calculated using Eq.(9) and (10), visibility data for other cases were obtained from PWD measurements."*

Because visibility observations were unavailable for fog cases 1-3 and 18-20, visibility for these cases was estimated using the observed fog droplet spectra and the microphysical parameterization of Gultepe (2006), which is based on the visible-light extinction theory in fog. The detailed formula and calculation procedure are provided in Section 2 *Data set and Methods* as follows:

*"Because visibility observations were unavailable for fog cases 1-3 and 18-20, visibility for these cases was estimated using the observed fog droplet spectra and the microphysical parameterization scheme developed by Gultepe (2006), based on the extinction theory of visible light in fog:*

$$V = \frac{-ln\alpha}{\beta_{ext}} \qquad (9)$$

*in which*

$$\beta_{ext} = \pi \sum Q_{ext} n(r) r^2 dr \qquad (10)$$

*where $\alpha$ is the constant threshold, typically set to 0.02, $\beta_{ext}$ represents the extinction coefficient, $Q_{ext}$ is the Mie extinction efficiency, which depends on particle radius, number concentration, and the wavelength of visible light. When droplet size exceeds about 4 µm, $Q_{ext}$ approaches a constant value of 2. For smaller droplets (less than 4µm), $Q_{ext}$ varies between 0.9 and 3.8 (Brenguier et al., 2000; KOENIG, 1971)."*

Q5- Fig. 1 summarized the results; when I see this figure, I thought this paper should be rejected. Reason is that single mode is very different than others but other are almost same values with very small differences.

Anwser: We appreciate this important concern. In the revised manuscript, we fitted the observed DSDs with unimodal, bimodal, and trimodal gamma and lognormal distributions. For each DSD, unimodal, bimodal, and trimodal fits using both the gamma and lognormal distribution were evaluated. The Akaike Information Criterion (AIC) and the Bayesian Information Criterion (BIC) were used to determine which of the three fits provides the best representation of the given DSD within each distribution type, and this fit was considered as the optimal fit. AIC provides a numerical basis for ranking competing models by their information loss in approximating the unknown true process, with the model yielding the lowest AIC considered the best approximating model (Symonds and Moussalli, 2011). BIC is consistent in the sense that it selects the true model with probability approaching one. A lower BIC corresponds to a higher posterior probability for the model and is therefore regarded as indicating a better model (Chakrabarti and Ghosh, 2011). Within each distribution type, the unimodal, bimodal, and trimodal fits are compared, and the one with the lowest AIC and BIC was considered the optimal fit of the given DSD.

We then retrieved $N_f$, $LWC$, $R_v$ and $R_{eff}$ from the optimal gamma and lognormal fits and compared them with the observations. The lognormal function provided a more accurate representation of the DSDs, so we used it to determine the number of peak modes and peak diameters in this study, and we reclassified all fog cases accordingly. The detailed formulas, calculation procedures, figures and descriptions of AIC and BIC are provided in Section 2 (*Data Set and Methods*). As a result in the revised manuscript, the classification of the fog events differs from that in the original submission. In the new results, Figure 1 has been updated and is now presented as Figure 3. For the unimodal mean DSD, the number concentration of droplets larger than 10 µm is lower than in the other cases and lower than the overall mean, as shown below:

[Figure]

**Figure 3 Average spectrums of fog events with different modes**

Q6- Fig. 1 and analysis; what are the time averages used? 1 sec? 1 min , 10 mins, 60 mins and why? I don't see this. Your results are time dependent/averages. All results should be revaluated.

Answer: Thank you for your suggestion. We have added a discussion on the 1-min and 5-min temporal resolutions at the end of Section 2 (*Data and Methods*) in the revised manuscript. We fitted the DSDs at both resolutions, derived the peak mode numbers and peak diameters, and reclassified all fog cases accordingly. The results show that the determination of peak mode numbers and peak diameters of DSD is not sensitive to temporal resolution, with only minor differences between the two. To reduce noise, we adopt the 5-min resolution in this study. The detailed classifications and modal diameters for both resolutions are provided in Table A2 in the appendix.

[Figure]

**Figure 2 (a) The $D_{peak}$ distributions of all DSDs at 1-min and 5-min resolution, and (b) the $D_{peak}$ distributions by fog type at both resolutions**

Q7- Where are the time series of Nd, LWC, and MVD for at least 2 events?

Answer: As recommended, the revised manuscript includes the temporal evolution of Nd, LWC, and MVD for the five fog cases in Figures 4-8, shown in subpanel (a) of each figure.

[Figure]

*Figure 4 (a) is the temporal evolution of $N_f$, LWC, $R_v$, $R_{eff}$, FBS and dispersion for fog case 22, the dark lines represent 5-minute averaged values while the light lines are 1-minute averaged values. (b) is the 5-minute average DSD. As LWC reaches its maximum, the colors vary from blue to red, and each DSD in (b) are marked by colored dots in (a) with each color corresponding to a specific DSD and its number of peaks indicated. (c) is the temporal evolution of visibility, temperature and wind speed, with wind direction represented by wind barbs. (d) is the 1-minute temporal evolution of DSDs.*

Q8- How come LWC becomes close to 1 g m-3? For some cases?

Anwser: Thank you for your question. In this study, the maximum LWC is approximately 0.8 g m⁻³, occurring mainly in cases with high droplet number concentrations such as Fog Case 1-1. Similar LWC values close to 1 g m⁻³ have also been reported in another winter fog study in Nanjing (Wang Y, Lu C, Niu S, et al. Diverse dispersion effects and parameterization of relative dispersion in urban fog in eastern China[J]. Journal of Geophysical Research: Atmospheres, 2023, 128(6): e2022JD037514.).

[Figure]

*Figure 7. Same with Figure 5 but for the fourth fog case.*

*(Figure 5. Time series of (a/b/c) fog droplet size distribution (FDSD), (d/e/f) number concentration ($N_f$), liquid water content (LWC), (g/h/i) volume-mean diameter ($D_v$), relative dispersion ($\varepsilon$), (j/k/l) the slope of the moving linear fit for continuous five $\varepsilon$ versus Dv points (slope), and the first bin strength (FBS) during the three fragments with different developing stages in the first fog case.)*

Q9- Nd is more than 200 cm-3 and goes up to 600 cm-3; where is PWD Vis comparison?

Answer: Thank you for your suggestion. Visibility data for all fog cases have been added to Table A1 in the appendix, including the mean, 25th percentile, and 75th percentile. For the five fog events analyzed in detail (fog case 1-1, fog case 10, fog case 4, fog case 22 and fog case 20), visibility time series have also been included. As shown in the Figure 4-8, visibility varies in close correspondence with $N_d$. For example:

[Figure]

Figure 4 (a) is the temporal evolution of $N_f$, LWC, $R_v$, $R_{eff}$, FBS and dispersion for fog case 22, the dark lines represent 5-minute averaged values while the light lines are 1-minute averaged values. (b) is the 5-minute average DSD. As LWC reaches its maximum, the colors vary from blue to red, and each DSD in (b) are marked by colored dots in (a) with each color corresponding to a specific DSD and its number of peaks indicated. (c) is the temporal evolution of visibility, temperature and wind speed, with wind direction represented by wind barbs. (d) is the 1-minute temporal evolution of DSDs.

Q10- At least show a couple picts from camera for dense fog events?/satellite images please, 3.9 micron channels.

Answer: We appreciate your suggestion. For F4 and F22, no satellite imagery is available within the fog period because the overpass times of the polar-orbiting satellites did not coincide with the observations. In the revised manuscript, we have included MODIS Aqua and Terra imagery for F1-1, F10, and F20 in the 3.9 μm shortwave infrared and visible channels in the appendix (Figures A2-A4). For example:

[Figure]

Figure A2 Aqua MODIS 3.9 μm shortwave infrared and visible channel imagery for Fog 1-1, with the observation site marked by a red cross.

Q11- Ln27; fog definition is wrong, see Gultepe et al 2016 Gultepe et al Atmos Res; AMS Bull on ice fog. Fog can be liquid or ice …

Anwser: Thank you for pointing out this error. It was indeed an oversight on our part. It has now been corrected as follows:

*"Composed of suspended small water droplets or individual ice crystals in the air near the surface, fog has multiple impacts ranging from transportation, vegetation, air quality, human health and economy (Gultepe et al., 2014, Jia et al., 2019; Lakra and Avishek, 2022)."*

Answer: Thank you for this suggestion and for providing the reference. The relevant content has been added to the *Introduction* in the revised manuscript. The specific revisions are provided below:

*"The C-FOG field campaign along the Atlantic Canada and northeastern U.S. coastlines showed that coastal fog was influenced by multiple weather systems, including northeastern high pressure, west-northwest low pressure, and tropical cyclonic activity (Gultepe et al., 2021)."*

In addition, we included a new content discussing the influence of large-scale processes on fog in the same study region as the cited reference:

*"Another study in the region found that fog associated with cyclonic systems was consistently produced by cloud-base lowering and subsequent downward extension to the surface, whereas anticyclonic fog developed either from surface radiative cooling or from the downward extension of low-level stratus to the surface (Dorman et al., 2021)."*

Q13- Ln43,44,45 ; 3, 3,5 and 4.5 micron numbers mean nothing because they are in the range of measurement uncertainty at about 1-2 micron.

Answer: Thank you for the suggestion. The corresponding content in the original *Introduction* has been removed and replaced with references to other studies on the multimodal fog droplet size distribution, such as:

*"Gultepe and Milbrandt (2007) reported DSD modes near 4 and 23 µm during winter fog events in the Toronto region. Boudala et al. (2022) investigated the seasonal and microphysical characteristics of fog at Cold Lake Airport in northern Alberta, Canada, and found that radiation fog exhibited a bimodal droplet spectrum with peaks at 4 µm and 17-25 µm."*

Q14- First bin is removed (ref: Gultepe and Isaac QJRMS 200?)

Answer: Thank you for the suggestion. The current approach in our manuscript, which removes the first bin of the FM-100 data, follows the method used by Lu et al. in their study of fog in Nanjing, China in 2007. The specific reference is: *Lu, C., Liu, Y., Niu, S., Zhao, L., Yu, H., and Cheng, M.: Examination of microphysical relationships and corresponding microphysical processes in warm fogs, Acta Meteorol. Sin., 27, 832–848, https://doi.org/10.1007/s13351-013-0610-0, 2013.* And the relevant content in their paper is: *"the size distributions of fog droplets were measured with a droplet spectrometer (FM-100) (Eugster et al., 2006; Gultepe et al., 2009). It measures fog droplets of 0.5-25 µm in radius at a sampling rate of 1 Hz, but the data from the bin 0.5-1.0 µm are thought to be noisy, which are not included in the calculations of microphysical properties such as N, volume-mean radius ($r_v$), standard deviation (σ), and LWC."*

We have added this citation to the revised manuscript, as shown below:

*"To exclude the influence of large unactivated aerosol particles, data from the first bin (1-2 µm) are omitted (Lu et al., 2013)."*

Regarding the reference you suggested, the information of the publication year was not complete in the review comment. We therefore searched Google Scholar for all papers by Gultepe and Isaac published in QJRMS (Quarterly Journal of the Royal Meteorological Society) between 2000 and 2010. Two papers were found: *'An analysis of cloud droplet number concentration ($N_d$) for climate studies: Emphasis on constant $N_d$'* and *'Aircraft observations of cloud droplet number concentration: Implications for climate studies'*. The first paper is only cited by others and its full text was not available. In the second paper, we did not find any discussion related to

removing FM-100 bins. If you have other specific reference in mind, we would be grateful if you could let us know.

Q15- Ln 49; how do you know because of collision processes? Where is the wind measurements in this paper?

Answer: Line 49 "When the fog DSD is bimodal, there is a mass transfer from smaller droplets to larger droplets due to collision-coalescence process, while sedimentation by gravity speeds up the removal of fog droplets." This statement is based on the conclusions of Mazoyer et al. (2022). To make the source explicit, we have added the citation again at the end of the sentence, revised as follows:

*"When the fog DSD is bimodal, there is a mass transfer from smaller droplets to larger droplets which may due to collision-coalescence process, while sedimentation by gravity speeds up the removal of fog droplets (Mazoyer et al., 2022)."*

In the revised manuscript, wind speed and wind direction data have been added to Table A1 in the appendix and to Figures 4-8 in the main text. We have also added an analysis of the influence of wind on fog in Section 3.2 *Mode Transitions and Possible Mechanisms*, for example:

"*Fog Case 10 (F10) formed as a radiation fog under high-pressure control and weak surface winds (Figure A9). and remained stable with relatively low $N_f$ and LWC for the first 6 hours. After 2007/12/20 06:00, rising temperature and variable wind direction enhanced turbulent mixing and promoted fog development. $N_f$ and LWC increased with pronounced fluctuations, which were also reflected in the evolution of the DSD.*"

Answer: There was indeed an error here, and we greatly appreciate your correction. The text has been revised to:

*"A simulation of a heavy fog event in North China Plain found that effective radius of fog droplet decreases nonlinearly with aerosol number concentration (Jia et al., 2019)."*

Q17- Ln59; T and RH, wind measurements still have very large uncertainty in predictions; who said they are correct?

Answer: Thank you for pointing this out. The inappropriate wording in the original manuscript has been revised to:

"*Despite WRF has made progress in forecasting certain variables such as temperature and wind, it often struggles to capture the accurate fog lifecycle (Peterka et al., 2024; Román-Cascón et al., 2016).*"

Answer: Thank you for your correction. The text has been revised to:

*"Currently, fog DSDs are described using various spectral distribution functions such as exponential, gamma or lognormal functions in bulk and bin microphysical scheme (Kessler, 1969; Khain et al., 2015)."*

Q19- Ln73; what kind of fog; radiation? Advection? Marine etc.

Answer: Thank you for your suggestion. The classifications of the 27 fog events have been added to Table A1 in the appendix, including radiation fog, advection fog, and radiation-advection fog. For the five cases analyzed in detail, synoptic background analyses have also been added to the main text for example:

*"Fog Case1 was a radiation-advection fog event lasting 39 hours. It formed under radiative cooling conditions, with sustained southwest warm and moist airflow supporting its long duration (Figure A8)."*

Answer: Thank you for your correction. The text has been revised to:

*"The DSD was measured with a fog monitor (FM-100) from Droplet Measurement Technologies (DMT, USA) with diameters ranging from 1 to 50 μm into 20 bins, at a sampling frequency of 1 Hz."*

Q21- Method section is not clear to me, providing eqs are not enough. How did you make averages? Where is the Vis etc.

Answer: Thank you for your suggestion. We have added a discussion on the 1-min and 5-min temporal resolutions at the end of Section 2 (*Data and Methods*) in the revised manuscript. We fitted the DSDs at both resolutions, derived the peak mode numbers and peak diameters, and reclassified all fog cases accordingly. The results show that the determination of peak mode numbers and peak diameters of DSD is not sensitive to temporal resolution, with only minor differences between the two. To reduce noise, we adopt the 5-min resolution in this study. The detailed classifications and modal diameters for both resolutions are provided in Table A2 in the appendix.

[Figure]

**Figure 2 (a) The $D_{peak}$ distributions of all DSDs at 1-min and 5-min resolution, and (b) the $D_{peak}$ distributions by fog type at both resolutions**

Visibility data have been added to Table A1 in the appendix and to Figures 4-8 in the main text.

Answer: Thank you for your suggestion. For the 5 cases discussed in detail (F1-1, F4, F10, F20, and F22), the time series of DSDs at 1s resolution has been added to the appendix as Figure A11. For example:
"

[Figure]

**Figure A11 The time series of DSDs at 1s resolution of Fog Case 22, Fog Case 4, Fog Case 1-1, Fog Case 10 and Fog Case 20**"

Answer: Thank you for your question. As mentioned in Question 5, because the revised manuscript adopts a different method for determining the number of peak modes and their peak diameters, the results show slight changes, and the phenomenon "*Note that the mean DSD of trimodal events shows higher $N_f$ than that of bimodal cases in the 10-35 μm range, but lower $N_f$ beyond 35 μm.*" described in the original version is no longer evident. In the revised analysis, for droplets larger than 20 μm, the mean spectra of the trimodal cases and of all cases combined exhibit significantly higher number concentrations than the unimodal and bimodal

cases, indicating that large droplets are primarily contributed by the trimodal cases. This may be because, in this study, trimodal cases are often associated with radiation-advection fog, where sustained warm and moist inflow allows the fog to persist longer and facilitates the growth of droplets to larger sizes. In the revised manuscript, Figure 1 have been updated to Figure 3 and the corresponding text have been updated as follows:

*"Figure 3 shows the mean DSDs for fog events with different modes, as well as the overall mean DSD across all cases. For droplet sizes smaller than 20 μm, the spectral distributions across cases with different modes are similar, particularly for droplets below 6 μm. And the mean spectrum closely matching that of the trimodal cases. As aerosol activation is governed by environmental supersaturation and the aerosols hygroscopic properties (Shen et al., 2018; Wang et al., 2019), the similar $N_f$ at the small-droplet end may suggest that aerosols continue to activate throughout fog development in both unimodal and multimodal cases. For diameters above 20 μm, the mean $N_f$ in single-mode and double-mode cases decrease rapidly. Above approximately 35 μm, double-mode cases exhibit the lowest mean $N_f$. Triple-mode events consistently exhibit the broadest DSD and highest $N_f$ of large droplets, consistent with their higher LWC and greater dispersion noted earlier.*

*In the following section, four representative cases are selected to examine the lifecycle characteristics of fog events with different modes and the evolution of DSDs throughout the stages of fog formation, development, and dissipation."*

[Figure]

***Figure 3 Average spectrums of fog events with different modes***

Q24- figure 2; make a square box for DSD same length in x and y.

Answer: Thank you for raising this issue. Following your suggestion, we have modified the DSD plots so that the x- and y-axes have equal scales, and we adjusted the proportions of the three related panels to ensure a more balanced and coherent visual layout. The revised figures are shown in Figures 4-8 of the revised manuscript. For example:

[Figure]

*Figure 4 (a) is the temporal evolution of $N_f$, LWC, $R_v$, $R_{eff}$, FBS and dispersion for fog case 22, the dark lines represent 5-minute averaged values while the light lines are 1-minute averaged values. (b) is the 5-minute average DSD. As LWC reaches its maximum, the colors vary from blue to red, and each DSD in (b) are marked by colored dots in (a) with each color corresponding to a specific DSD and its number of peaks indicated. (c) is the temporal evolution of visibility, temperature and wind speed, with wind direction represented by wind barbs. (d) is the 1-minute temporal evolution of DSDs.*

Q25- Until fig 2; no averages mentioned; why those averages are taking?

Answer: Thank you for your suggestion. We have added a discussion on the 1-min and 5-min temporal resolutions at the end of Section 2 (*Data and Methods*) in the revised manuscript. We fitted the DSDs at both resolutions, derived the peak mode numbers and peak diameters, and reclassified all fog cases accordingly. The results show that the determination of peak mode numbers and peak diameters of DSD is not sensitive to temporal resolution, with only minor differences between the two. To reduce noise, we adopt the 5-min resolution in this study. The detailed classifications and modal diameters for both resolutions are provided in Table A2 in the appendix.

[Figure]

*Figure 2 (a) The $D_{peak}$ distributions of all DSDs at 1-min and 5-min resolution, and (b) the $D_{peak}$ distributions by fog type at both resolutions*

Q26- Fig 3a Nf spectra, pdf are almost same; is it contradict to other events/cases etc. what is averaging time?

Answer: Thank you for your question. In the revised manuscript, as mentioned in question 5, we use lognormal

fitting to determine the DSD peak diameters and the number of peak modes, which leads to some differences compared with the original method based on identifying local minima in the DSDs. Figure 3a has been updated to Figure 9a, where the PDF of unimodal DSDs at Nf > 250 cm$^{-3}$ is much lower than that of the other DSD types, consistent with Figure 3. The specific revisions are as follows:

"

[Figure]

*Figure 9 PDF distributions of $N_f$ (a), LWC (b), $R_v$(c) and $R_{eff}$ (d) for DSDs with different modes"*

Q27- Fig06; represents what? Using obs or theretical data?

Answer: Thank you for your question. In the revised manuscript, Figure 6 has been revised as Figure 16, which now shows the correlations among the gamma fit parameters under different segmentations, reflecting the characteristics of the spectral shape. In the Gamma distribution, $N_0$ describes the overall magnitude of the droplet number concentration, while the shape parameter $\mu$ indicates whether the spectrum is concave or convex, and the slope parameter $\lambda$ controls the rate at which number concentration decreases with increasing droplet size, thus determining the spectrum width (Lee et al., 2023). Therefore, analyzing the correlations among Gamma parameters provides physical insight into the droplet size distribution evolution across different droplet size ranges. The negative correlations of $N_0$ with $\mu$ and $\lambda$ indicate that higher concentrations tend to be associated with spectra dominated by smaller droplets and broader size distribution. The positive $\mu$-$\lambda$ relationship suggests that when the spectrum has larger peak diameter, the number concentration decreases more rapidly with droplet size, and the spectrum becomes narrower in large-droplet ranges.

Q28- Eq. 11; not z but dz

Answer: Thank you for your correction. In the revised manuscript, Eq. 11 has been renumbered as Eq. 17, and the updated formula is as follows:

$$\text{``} \frac{\tau}{dz} = \frac{3LWC}{D_{eff}} \text{''}$$

Q29- Fig. 7; you have many points with LWC>0.5 g m-3? In polluted environment you should have that high. Why is that?

Answer: We appreciate your comment. Across all LWC data, values greater than 0.5 g m$^{-3}$ account for only about

2%, occurring infrequently and not in continuous periods, mainly in stronger fog events such as fog case 1. Similar LWC values above 0.5 g m⁻³ have also been reported in another winter fog study in Nanjing (Wang Y, Lu C, Niu S, et al. Diverse dispersion effects and parameterization of relative dispersion in urban fog in eastern China[J]. Journal of Geophysical Research: Atmospheres, 2023, 128(6): e2022JD037514.).

Q30- LN315-318; see rev paper on Fog Pure and Appl Geop 2007 Gultepe et al.

Answer: Thank you for providing the reference. Following the suggestion from another reviewer, we have removed the *Discussion* section (the lines you mentioned, 315-318, were part of the *Discussion* section). However, we have incorporated the reference you suggested into the relevant part of the *Introduction*. The specific revision is as follows:

"*However, the fog DSD exhibits strong spatial and temporal variability and evolves throughout the fog lifecycle, often displaying distinct features and deviating from idealized distributions due to turbulent mixing, radiative effects, and gravitational settling (Gultepe et al., 2007; Nelli et al., 2024; Tampieri and Tomasi, 1976; Wang et al., 2021).*"

Q31- Ln342; I don't think Nf increases with broadening, should decrease. This contradicts everything I know for DSD. You need wind measurements…….

Answer: Thank you for your suggestion. The original statement "*$N_f$ increases and DSD broadens as fog develops while DSD transitions from multimodal to unimodal during dissipation*" was intended to describe the following: at the onset of fog, only small droplets are present (as highlighted in the red box below), with maximum diameters of about 30 μm and a relatively narrow spectrum. As the fog develops toward its mature stage, droplet number concentrations increase across all size ranges, and continued condensational growth leads to the appearance of droplets larger than 30 μm. In other words, both $N_f$ and spectral broadens as the fog develops.

[Figure]

To avoid ambiguity, the revised manuscript now states the following:

"*Among all fog cases, radiation fog accounts for the largest proportion. Radiation-advection fog tends to persist longer and is typically associated with trimodal DSDs. Unimodal cases are more likely to occur when the fog duration is short, the DSD is narrow, or the FBS is high. For bimodal and trimodal cases, both the number of*

*peaks and their diameters vary with the fog life cycle. As the fog develops, sustained condensational growth often leads unimodal and trimodal DSDs to evolve into bimodal, with more concentrated peaks. The peak diameters are linked to the ability of fog to maintain high number concentrations and liquid water content. Although the $N_f$ of larger droplets increases, droplets around $3\,\mu m$ consistently exhibit the highest $N_f$, indicating continuous activation and formation of new droplets during the condensational growth."*

**Reference**

Lee, G., Bringi, V., and Thurai, M.: The Retrieval of Drop Size Distribution Parameters Using a Dual-Polarimetric Radar, Remote Sensing, 15, 1063, https://doi.org/10.3390/rs15041063, 2023.